# Integrated multi-modal data analysis for computational modeling of healthy and location-dependent myocardial infarction conditions in porcine hearts

Ricardo M. Rosales[1,2,3*], Ming Wu[4], Piet Claus[4], Stefan Janssens[4], Gonzalo R. Ríos-Muñoz[5,6,7], María Eugenia Fernández-Santos[6,7], Pablo Martínez-Legazpi[6,7,8], Javier Bermejo[6,7,9], Aiden Flanagan[10], Manuel Doblaré[1,2,3,11], Ana Mincholé[1,2,3,11], Esther Pueyo[1,2,3,11]

**1** Aragón Institute for Health Research (IISA), Zaragoza, Aragón, Spain, **2** Aragón Institute of Engineering Research (I3A), Zaragoza, Aragón, Spain, **3** University of Zaragoza, Zaragoza, Aragón, Spain, **4** Department of Cardiovascular Sciences, University of Leuven, Leuven, Belgium, **5** Bioengineering Department, Universidad Carlos III de Madrid, Leganés, Spain, **6** Department of Cardiology, Instituto de Investigación Sanitaria Gregorio Marañón (IiSGM), Hospital General Universitario Gregorio Marañón, Madrid, Spain, **7** Center for Biomedical Research in Cardiovascular Disease Network (CIBERCV), Madrid, Spain, **8** Department of Mathematical Physics and Fluids, Universidad Nacional de Educación a Distancia (UNED), Madrid, Spain, **9** Departamento de Medicina, Universidad Complutense, Madrid, Spain, **10** Boston Scientific Limited, Galway, Ireland, **11** CIBER-BBN, Instituto de Salud Carlos III, Madrid, Spain

* rrosales@unizar.es

## Abstract

Porcine hearts are widely used for preclinical cardiac evaluation. Computational models, by effectively integrating comprehensive experimental data, often reinforce this preclinical assessment. Using extensive multi-modal data, we developed swine ventricular digital twins for healthy and chronic myocardial infarction (MI) conditions to investigate the roles of the cardiac conduction system (CS), spatial repolarization heterogeneities, cardiomyocyte orientation, cell-to-cell coupling, and MI characteristics on ventricular function. We analyzed cardiac magnetic resonance (CMR) images, electrocardiograms (ECGs), and optical (OM) and electroanatomical mapping from 5 healthy and 10 MI pigs. CS architectures were built from OM and ECG recordings. Myocardial fiber orientation, action potential characteristics, and cell-to-cell conductivity in MI tissue were defined from OM and CMR data. Simulated ECGs for healthy and MI models of left anterior descending and left circumflex occlusions were compared to experimental ECGs and were used to assess MI-induced changes. Subject-specific fiber orientation calibration minimally affected electrophysiology, with conduction velocity (CV) and action potential duration (APD) changing by less than 3.6% with respect to standard orientation. Accurate CS and repolarization heterogeneities reproduced depolarization (Pearson correlation 0.76 for QRS) and repolarization (Pearson correlation 0.74 for T-wave) patterns. Incorporating experimentally guided MI-induced alterations enabled the replication of MI depolarization

**Data availability statement:** The modeling and postprocessing software are publicly available at https://github.com/lino202/HeartModelling and https://github.com/lino202/simOmPP, while the simulation solver is provided at https://github.com/lino202/ELECTRA. The computational models and simulation configuration files are available at https://doi.org/10.5281/zenodo.17415592, while the experimental data are available at https://doi.org/10.5281/zenodo.18223187.

**Funding:** RMR, MW, PC, SJ, GRRM, MEFS, PML, AF, MD, and EP received financial support from the EU H2020 Program under G.A. 874827 (BRAV3). RMR, AM, and EP were supported by Agencia Estatal de Investigación - Ministerio de Ciencia e Innovación (Spain) through projects PID2022-140556OB-I00, PID2023-148975OB-I00, TED2021-130459B-I00, and CNS2022-135899, by Aragón Government through BSICoS group T39_23R, and by the European Research Council under G.A. 638284. RMR, MD, and EP were supported by Agencia Estatal de Investigación - Ministerio de Ciencia e Innovación (Spain) through project CARDIOPRINT (PLEC2021-008127). GRRM was supported by Madrid Government (Comunidad de Madrid) under the Multiannual Agreement with UC3M (FLAMA-CM-UC3M), and through project MAGERIT-CM (TEC-2024/COM-44). The funders had no role in study design, data collection and analysis, decision to publish, or preparation of the manuscript.

**Competing interests:** I have read the journal's policy and the authors of this manuscript have the following competing interests: AF is employed by Boston Scientific Corporation and states: "The content of this publication is under the sole responsibility of its author/publisher and does not represent the views or opinions of Boston Scientific Corporation".

and repolarization features (relative errors: 0.5% CV, 2.9% APD), yielded realistic T-wave morphologies (0.63 Pearson correlation), and revealed ECG patterns specific to vessel-dependent occlusions. Thus, by integrating extensive multi-modal data, we advance porcine cardiac digital twins and demonstrate the influence of key structural and electrophysiological parameters on healthy and MI heart function, providing a robust computational framework for mechanistic and translational applications.

## Author summary

Pigs are commonly used as preclinical models for cardiac evaluation due to their close resemblance to the human heart. Computational cardiac electrophysiology often supports and extends this preclinical assessment. However, the reliability of these *in silico* representations depends on the effective integration of comprehensive experimental data. Using extensive multi-modal data, we developed swine ventricular digital twins under healthy and chronic myocardial infarction conditions. These models allowed us to investigate the influence of the cardiac conduction system, spatial repolarization heterogeneities, cardiomyocyte orientation, cell-to-cell coupling, and vessel-specific infarction characteristics on ventricular function. Our results show that cardiomyocyte orientation exerts only a minor effect on electrophysiology, with average conduction velocity showing a slight decrease and action potential duration remaining unchanged when comparing standard versus individualized orientations. Accurate representation of conduction system architecture and repolarization heterogeneities enabled close reproduction of experimental depolarization and repolarization patterns. Furthermore, by integrating experimentally guided reductions in cell-to-cell coupling and inward rectifier potassium current, along with individualized post-infarction activation alterations and a novel porcine cellular model, we were able to faithfully replicate infarction-specific depolarization-repolarization features. These refinements produced realistic T-wave morphologies and revealed electrocardiographic signatures associated with vessel-dependent infarctions, underscoring the translational potential of our approach.

## 1 Introduction

Ischemic heart disease remains the leading global cause of death in both men and women [1]. When an ischemic episode persists for several minutes, it can lead to myocardial infarction (MI), in which the necrosis of non-renewable cardiomyocytes unleashes an orchestrated inflammatory response that leads to the replacement of dead tissue with fibrotic connective tissue [2]. The infarction-affected zone (AZ) is usually divided into scar (SZ) and border (BZ) zones. SZ is a dense, non-excitable fibrotic scar, mainly composed of collagen and characterized by the thinning and increased stiffness of the ventricular wall. BZ represents a transition region from SZ

to the healthy zone (HZ) characterized by myocardial disorganization and increased collagen content [3,4]. Beyond these structural alterations, MI also remodels the cardiac conduction system (CS), modifying the density and spatial distribution of the Purkinje-muscular junctions (PMJs), as reported in histological studies of chronic MI [5].

Chronic MI-induced ventricular remodeling increases the risk of heart failure and the vulnerability to arrhythmias [1,6,7]. The severity of post-MI cardiomyopathy strongly depends on the infarct location, with relevant differences found between the occlusion of the left anterior descending (LAD) and the left circumflex (LCx) coronary arteries. LAD occlusion involves larger infarcted areas, resulting in greater remodeling and increased impairment of systolic function compared with LCx occlusion [7].

Animals are widely used as models in preclinical therapeutic evaluations and as donors for transgenic xenotransplantation of organs. In the context of cardiovascular diseases, large white domestic pigs (*Sus scrofa domestica*) are the gold-standard choice due to their high similarity to humans in terms of coronary circulation, hemodynamics, cardiac size, and electromechanical behavior [8]. To implement the 3Rs (refine, replace, and reduce) of animal experimentation, promoting the humane and ethical use of animals while supporting research on the prophylaxis, diagnosis, and treatment of cardiovascular diseases, swine computational models have emerged as powerful tools that complement *in vivo* and *ex vivo* studies [9,10].

In this context, *in silico* modeling and simulation of cardiac electrical activity provide a robust framework to investigate the mechanisms driving the function of the heart in health and disease, as well as in response to pharmacological and tissue engineering therapies, among others [11–13]. Initially, generic models with average electrophysiological parameters were used for general-purpose simulations. Advances in computational power, memory capacity, and access to biochemical and imaging data have enabled biophysically detailed mathematical simulations across a wide range of scenarios [14,15]. For pigs, the CESC10 [16] and CRT-EPiggy19 [17] challenges constituted the first endeavors to provide multi-modal experimental data for individualizing preclinical models and assessing their performance. These subject-specific models have been shown to, for example, better reproduce inter-individual variability in the generation of reentrant arrhythmias than generic models [9]. However, the creation of fully individualized digital twins involves time-consuming and technically complex experiments, often limiting the feasibility of parameter individualization. Current efforts focus on identifying and tailoring key electrophysiological parameters required to accurately replicate specific heart conditions [9,18–22].

Although *in silico* models are commonly used to support preclinical trials in pigs, many are still largely derived from non-porcine data, which limits mechanistic accuracy, predictive power, and the individualization of key simulation parameters. To overcome these limitations, we propose that specific cardiac conditions can be faithfully reproduced in pigs by tailoring critical modeling parameters, which are identified through extensive multi-modal porcine data and systematically validated with large-scale simulations.

In this context, this work has four main objectives: 1) to define a reproducible pipeline for the development and calibration of biventricular (BiV) electrophysiological models in healthy and MI pig hearts by processing cardiac signals and images from *in vivo* and *ex vivo* pig experiments; 2) to identify the critical parameters required to *in silico* reproduce both species-specific (porcine) and subject-specific (individual pig) electrical characteristics in healthy and MI conditions; 3) to provide openly available, state-of-the-art BiV porcine models for use in preclinical cardiovascular studies (e.g., drug cardiotoxicity, remuscularization assessment), thereby reducing reliance on animal experimentation; 4) to demonstrate the potential of these experimentally guided *in silico* models for mechanistic analysis of infarction-induced electrocardiogram (ECG) alterations across multiple infarct locations.

Beyond these technical objectives, the framework is designed to address key biological questions in both healthy and MI conditions. In the healthy state, we investigate: (i) how CS architecture and depth shape ventricular activation, and the functional implications of the deeper CS observed in pigs; (ii) the extent to which regional repolarization differences contribute to T-wave formation, assessing the presence of relatively limited yet existing repolarization heterogeneity in the

porcine heart; and (iii) how fiber orientation shapes depolarization and repolarization patterns at both tissue and organ scales.

In the MI state, we shed light on: (i) whether post-MI alterations in ionic currents, particularly the inward rectifier potassium current and fast sodium current, and reduced cell-to-cell coupling constitute the primary mechanisms driving the post-MI electrophysiological phenotype; and (ii) how MI-induced changes in activation translate into ECG alterations. By linking these infarct-induced modifications to observed ECG changes across different infarct locations and extents, we provide mechanistic insight into ECG variability that is difficult to obtain experimentally in large-animal models.

## 2 Methods

### 2.1 Experiments

**2.1.1 Ethics statement.** All animal procedures complied with Directive 2010/63/EU of the European Parliament on the protection of animals used for scientific purposes, and adhered to the animal welfare regulations applicable in each country where the experiments were performed. All procedures were approved by the relevant local regulatory authorities, as follows: at Katholieke Universiteit Leuven, by the Animal Ethics Committee of the Katholieke Universiteit Leuven (approval number: ECD P114/2020); at Servicio Madrileño de Salud, by the Consejería de Agricultura, Desarrollo Rural, Población y Territorio – Dirección General de Agricultura y Ganadería – Servicio de Sanidad Animal – Junta de Extremadura (approval number: EXP-20210628), and by the Dirección General de Agricultura, Ganadería y Alimentación – Consejería de Medio Ambiente, Ordenación del Territorio y Sostenibilidad – Comunidad de Madrid (approval number: PROEX 148.8/20).

**2.1.2 Animals.** The study population consisted of 15 large white domestic pigs (Fig 1a), of both sexes, sourced from two centers: Katholieke Universiteit Leuven (KUL), Leuven, Belgium, and Servicio Madrileño de Salud (SERMAS), Madrid,

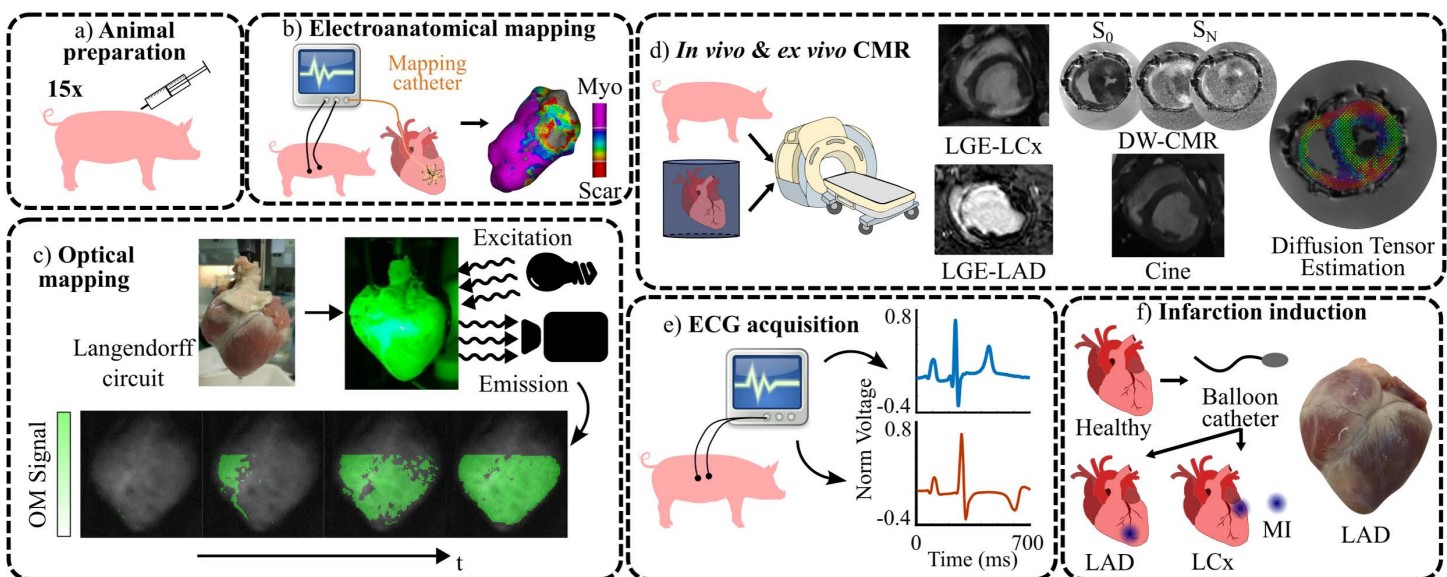

**Fig 1. Experimental data acquisition.** a- Animal preparation was conducted on 15 pigs. b- EAM of pig 6. c- OM scheme with activation of pig 13. d- Different CMR techniques applied. LGE-LCx: pig 7, LGE-LAD: pig 10, cine: pig 3, DW-CMR and diffusion tensor estimation: pig 2. e- ECG acquisition. It shows the ECG signal at lead II for healthy pig 15 (blue) and LCx-MI pig 6 (orange). f- Scheme of MI induction with a picture of the anterior view of an LAD-MI swine heart. Healthy data was acquired prior to MI induction. MI data was acquired 4 weeks after MI (chronic stage). EAM: Electroanatomical mapping, OM: Optical mapping, CMR: Cardiac magnetic resonance, LGE: Late gadolinium-enhanced, DW-CMR: Diffusion-weighted CMR, LCx: Left circumflex infarction, LAD: Left anterior descending infarction, ECG: Electrocardiogram, MI: Myocardial infarction.

Spain. Animals weighing 20–30 kg (KUL) and 55–65 kg (SERMAS) were included. Structural and functional data from both healthy and MI pigs were collected and processed as detailed in Table 1.

Prior to each experiment, the animals were sedated and anesthetized following site-specific protocols. At KUL, animals were sedated with Telazol (tiletamine 4 mg/kg and zolazepam 4 mg/kg, Zoletil100, Virbac Animal Health, Carros, France) combined with xylazine (2.5 mg/kg, Vexylan, CEVA Sante Animale, Brussels, Belgium). Anesthesia was induced intravenously with propovet (10 mg/kg, propofol, Med'Vet, Chatillon, France), followed by continuous infusion of the same anesthetic (10 mg/kg/h) and remifentanil (18 μg/kg/h, Ultiva, GSK, Genval, Belgium). At SERMAS, a standardized anesthesia protocol was applied. Premedication was administered intramuscularly using ketamine (10 mg/kg), midazolam (0.3 mg/kg), and atropine (0.05 mg/kg). Anesthesia was induced intravenously with fentanyl (0.01 mg/kg) and propofol (4 mg/kg). Maintenance of anesthesia was sustained by continuous intravenous administration of fentanyl (0.01 mg/kg) and propofol (14 mg/kg).

**2.1.3 Induction of myocardial infarction.** LCx artery MI and LAD artery MI were induced at KUL and SERMAS, respectively (Table 1 and Fig 1f). At KUL, pigs received amiodarone orally for two weeks (400 mg/d during the first week, 200 mg/d during the second week). One day before MI induction, animals were preloaded with acetylsalicylic acid and clopidogrel (300 mg each) via oral gavage. Sedation and anesthesia protocols followed those described in Section 2.1.2. Mechanical ventilation with an air-oxygen mixture (1:1) was adjusted to maintain normocapnia and normoxia using a tidal volume of 8–10 mL/kg. Continuous monitoring of blood pressure, ECG, and peripheral oxygen saturation was performed using an intensive cardiac monitor system (Siemens). Following anticoagulation with heparin (10,000 IU) and antiplatelet administration (acetylsalicylic acid, 500 mg), MI was induced by proximal occlusion of the LCx coronary artery. Briefly, a guide wire was advanced through the coronary artery, and a dilation balloon catheter, sized according to angiographic measurements, was positioned and inflated for 90 min to occlude the vessel, followed by deflation to allow reperfusion. Coronary artery occlusion and reperfusion were confirmed by angiography. At SERMAS, MI was induced by balloon occlusion of the mid-LAD coronary artery, distal to the first diagonal branch, via femoral access. Occlusion was maintained for 90–150 min to achieve TIMI 0 flow, followed by reperfusion. Lidocaine (1 mg/kg IV) was administered prior to

**Table 1. Summary of collected and processed experimental data. HE: Healthy, LCx: Left circumflex infarction, LAD: Left anterior descending infarction, KUL: Katholieke Universiteit Leuven, SERMAS: Servicio Madrileño de Salud, MR: Magnetic resonance, LGE: Late gadolinium-enhanced, DW: Diffusion-weighted, OM: Optical mapping, EAM: Electroanatomical mapping, ECG: Electrocardiogram.**

| Pig | State | Center | MR | | | | Signals | | |
|---|---|---|---|---|---|---|---|---|---|
| | | | Cine | LGE | DW | Thoracic | OM | EAM | ECG |
| 1 | HE | KUL | X | | X | | | | |
| 2 | HE | KUL | X | | X | | | | |
| 3 | HE | KUL | X | | X | | | | |
| 4 | LCx | KUL | | X | | | | | |
| 5 | LCx | KUL | | X | | | | | |
| 6 | LCx | KUL | | X | | X | | X | X |
| 7 | LCx | KUL | | X | X | | | | |
| 8 | LAD | SERMAS | | X | | | X | | |
| 9 | LAD | SERMAS | | X | | | X | | |
| 10 | LAD | SERMAS | | X | | | X | | |
| 11 | LAD | SERMAS | | X | | | X | | |
| 12 | LAD | SERMAS | | X | | | X | | |
| 13 | HE | SERMAS | | | | | X | | |
| 14 | LAD | SERMAS | | | X | | | | |
| 15 | HE | KUL | | | | | | | X |

occlusion to prevent arrhythmias. Ventricular tachyarrhythmias were treated according to standard resuscitation protocols. Procedures were conducted under general anesthesia with sevoflurane and mechanical ventilation via orotracheal intubation. After MI, pigs received aspirin (500 mg QD), clopidogrel (300 mg QD), and amiodarone (400 mg QD for 5 days). This reperfusion model has been previously validated and produces a heterogeneous scar that more accurately represents clinical MI cases [23,24]. MI data were collected 4 weeks after MI induction (chronic stage).

## 2.2 Cardiac magnetic resonance

The cardiac magnetic resonance (CMR) sequences analyzed in this study included cine-CMR, diffusion-weighted CMR (DW-CMR), and late gadolinium-enhanced CMR (LGE-CMR) (Fig 1d). At KUL, all CMR acquisitions were performed during suspended respiration using electrocardiographic triggering and cardiac-dedicated surface coils on a 3T Siemens system (TRIO-Tim, Siemens, Erlangen). At SERMAS, DW-CMR and LGE-CMR were performed using 1.5T Philips Achieva and Intera magnetic resonators (Philips Healthcare, Best, The Netherlands).

**2.2.1 Cine-CMR.** Cine-CMR was performed to capture the diastole-systole motion of the heart by acquiring images of cardiac slices throughout the cardiac cycle. Following sedation and general anesthesia, mechanical ventilation was applied with an air-oxygen mixture (1:1) at a tidal volume of 8–10 mL/kg.

The three collected cine-CMR datasets had an anisotropic voxel resolution of 1.33 x 1.33 x 6 mm$^3$. End-diastolic volumes were selected for analysis. Preprocessing included cropping, padding, anisotropic resampling, and Z-score intensity normalization. Based on our previous work [25], the blood pools of the left and right ventricles (LV and RV) and the LV myocardium were segmented using a semi-automatic, deep learning-based pipeline that combined synthetic data augmentation and topological correction.

**2.2.2 LGE-CMR.** LGE-CMR was used to delineate MI by exploiting the preferential accumulation of gadolinium in AZ due to its easier penetration and delayed washout caused by the larger extracellular space between collagen fibers. The resulting higher signal intensity in T1-weighted CMR sequences allowed precise MI identification [26]. During acquisition, sedation, mechanical ventilation, and continuous vital sign monitoring were performed. At KUL, LGE-CMR was acquired 12 min after intravenous injection of gadolinium 0.2 mm/kg (Dotarem, Guerbet, Roissy, France). At SERMAS, LGE-CMR images were acquired 10 min after the injection of gadodiamide 0.2 mm/kg.

LGE-CMR data from SERMAS and KUL had anisotropic voxel resolutions of 1.28 x 1.28 x 2.5 mm$^3$ and 1.25 x 1.25 x 5.5 mm$^3$, respectively. Preprocessing and deep learning-based segmentation followed the same protocol as for cine-CMR data, with additional segmentation of the MI region [25].

**2.2.3 DW-CMR.** DW-CMR was used to characterize the three-dimensional (3D) microarchitecture of ventricular tissue by quantifying the directional diffusion rate of water molecules under applied electromagnetic gradients [27]. At KUL, hearts were arrested in end-diastole, excised with the great vessels, and perfused via the coronary arteries with 1 L of cold cardioplegic solution at a pressure of 100 mmHg, then immersed in the same solution for 1 hour. The great vessels were firmly tied to maintain pressure, and the hearts were placed in a individualized 3D-printed cast filled with cardioplegic solution (left panel of S1 Fig). These custom-designed containers were constructed using the epicardial segmentation of the *in-vivo* cine CMRs [28]. At SERMAS, an atrial transseptal puncture was performed to equalize intracavitary volumes. Warm 2% agarose gel solution (≈38 °C) was infused via cannulas in the aorta and pulmonary artery until the ventricular cavities approximated an *in vivo* diastolic state. The heart was positioned within a container of sufficient size to prevent compression and was embedded in warm agarose. Following embedding, the assembly was refrigerated to promote agarose gelation and stabilize the organ, thereby minimizing unwanted myocardial deformation and maintaining structural integrity (right panel of S1 Fig).

Multiple electromagnetic sensitizing gradients (Table 2) were applied to estimate myocardial diffusion tensors. Following segmentation, diffusion tensors were computed, and their primary eigenvectors were extracted to determine myocardial fiber orientation [29]. In brief, signal attenuation due to diffusion is modeled from DW-CMR using the Stejskal-Tanner

**Table 2. DW-CMR characteristics. DW-CMR: Diffusion-weighted cardiac magnetic resonance, HE: Healthy, LCx: Left circumflex infarction, LAD: Left anterior descending infarction.**

| Pig | State | Resolution (mm³) | Gradients | b value |
|---|---|---|---|---|
| 1 | HE | 1.09 x 1.09 x 1 | 64 | 1000 |
| 2 | HE | 1.14 x 1.14 x 1.2 | 48 | 1000 |
| 3 | HE | 0.98 x 0.98 x 1.3 | 36 | 1000 |
| 7 | LCx | 0.98 x 0.98 x 1.08 | 36 | 1000 |
| 14 | LAD | 1.09 x 1.09 x 1.2 | 15 | 600 |

equation: $S/S_0 = e^{-b\,d(g)}$, where $S/S_0$ represents signal attenuation, $d(g)$ is diffusion as a function of gradient, $b$ is the diffusion-weighting factor, and $S_0$ is the signal without the sensitizing gradients. Diffusion is modeled using a positive-valued Cartesian tensor of even order and full symmetry [30], where the primary eigenvector of this tensor corresponds to the longitudinal axis of the myocardial fibers at each spatial location [31].

In MI cases, AZ was manually added to the segmentation, and voxel-wise fractional anisotropy was calculated for HZ and AZ as a function of the diffusion tensor eigenvalues $\lambda_i$, $i = 1, 2, 3$:

$$\text{Fractional Anisotropy} = \sqrt{\frac{1}{2}\left(\frac{(\lambda_1 - \lambda_2)^2 + (\lambda_3 - \lambda_2)^2 + (\lambda_3 - \lambda_1)^2}{\lambda_1^2 + \lambda_2^2 + \lambda_3^2}\right)}$$

## 2.3 Optical mapping

Optical mapping (OM) was performed to record the electrical activity of explanted hearts (Fig 1c), using voltage-sensitive dyes to track changes in transmembrane voltage ($V_m$) with high spatio-temporal resolution [32].

Immediately after explantation, the hearts were immersed in cold (4 °C) cardioplegic solution containing (in mm): 12.6 NaHCO$_3$, 13.44 KCl, 280 D-glucose, 34 mannitol, 140 NaCl, and 10 2,3-butanedione monoxime, pH 7.4, for preservation during transport. On arrival, a cannula was inserted into the aorta to perfuse the heart via a Langendorff circuit with Tyrode's solution at 37 °C. The Tyrode composition was (in mm): 130 NaCl, 24 NaHCO$_3$, 4 KCl, 1 MgCl$_2$, 5.6 D-glucose, 1.2 NaH$_2$PO$_4$, and 1.86 CaCl$_2$, pH 7.4. Temperature (37 °C) and oxygenation were maintained via an external heating bath and an oxygenator with a carbogen gas mixture, using bubble traps and filters to ensure stable perfusion.

Electrical activity was monitored from fluorescence changes after infusion of a 100 μL bolus of di-4-ANEPPS (excitation: 482 nm, emission: 686 nm; Biotium, Inc. Hayward, CA, USA) in 4.16 mm DMSO applied for 5 min following electromechanical uncoupling with 10 μm 2,3-butanedione monoxime in Tyrode's solution. For dye excitation, hearts were homogeneously illuminated with a filtered green LED light source (LED: CBT-90-G; peak power output 58 W; peak wavelength 524 nm; Luminus Devices, Billerica, USA) through a planar-convex lens (LA1951; focal length 25.4 mm; Thorlabs, New Jersey, USA) and a green excitation filter (D540/25X; Chroma Technology, Bellows Falls, USA). Fluorescence was recorded using an EMCCD camera (Evolve-128: 128 × 128 imaging pixels, 24 × 24-um pixels, 16 bits; Photometrics, Tucson, AZ, USA) with a custom multiband emission filter (ET585/50–800/200 M; Chroma Technology) and high-speed lens (DO2595; Navitar Inc., Rochester, USA).

OM was performed on five porcine hearts with LAD MI and one healthy heart during sinus rhythm or under external stimulation applied from a catheter located in the right atrium, RV, or LV with a pacing cycle length of 1000 ms. Pixel resolution was determined from a conversion factor that was calibrated immediately before the OM experiment and held constant for all recordings (pig 8: 0.081, pig 9: 0.0729, pig 10: 0.0860, pig 11: 0.0885, pig 12: 0.0906, pig 13: 0.105 cm per pixel). The sampling frequency was 500 Hz in all cases.

Data were processed using custom software [33]. A region of interest that included both ventricles was manually defined [34]. High-pass filtering (0.4 Hz cut-off frequency) and adaptive spatio-temporal Gaussian filtering were applied to remove baseline drift and high-frequency noise.

The activation time (AT) for each pixel and beat was defined as the time interval from a fixed time reference to the time corresponding to the maximum positive derivative of $V_m$. Action potential (AP) duration (APD) at 90% repolarization ($APD_{90}$) was calculated as the difference between the time associated with the maximum derivative of $V_m$ and the time when 90% repolarization from the peak of the AP to the diastolic membrane potential was reached. Median AT and $APD_{90}$ values across beats were used to generate AT and $APD_{90}$ maps. Despite selecting consecutive beats with identical beating rates to minimize beat-to-beat variability in depolarization and repolarization measurements, residual variability remained. This was particularly evident in repolarization times near the infarct and in regions with significant fat coverage, such as the interventricular septum. For example, in pig 13 (n = 11 beats, sinus rhythm), the median standard deviation across pixels was 20.5 and 51.5 ms for ATs and $APD_{90}$ values, respectively, whereas in pig 10 (n = 4 beats, sinus rhythm), the corresponding values were 4.8 and 4.6 ms for ATs and $APD_{90}$ values, respectively. Given this variability, median values were considered a more robust and representative summary of the OM data. This level of experimental uncertainty, which can reach several tens of milliseconds, defines a realistic limit for parameter tuning. Furthermore, the conduction velocity (CV) vector between two pixels was defined as the ratio of the Euclidean distance between them to their AT difference. The mean CV vector for a given pixel was then calculated across its neighboring pixels within a radius of 3 pixels. The radius value was chosen as a trade-off to avoid over smoothed CVs while preventing extremely large CVs computed from noisy neighbors.

## 2.4 Electrocardiography

A 12-lead ECG was acquired from healthy pig 15 at KUL at a sampling frequency of 2000 Hz (Fig 1e). Lead placement was as described in Section 2.5. Signals were filtered using high- and low-pass Butterworth filters with cutoff frequencies of 0.5 and 40 Hz, respectively. ECG waves were delineated using a wavelet transform-based delineator [35]. From the detected QRS complexes, the mean RR interval was 769 ms.

To obtain a median beat $\overline{b}_i$ for each lead $i$, all beats ($b_i^j$, $j = 1, ..., N$, with $N$ representing the total number of beats in lead $i$) were aligned after automatically cropping them at 240 and 538 ms before and after the QRS fiducial point, respectively. The most representative beat $\hat{b}_i$ in lead $i$ was selected as the one with the highest sum of correlation coefficients with all other beats in the lead. The median beat $\overline{b}_i$ was then defined as the median of all $b_i^j$ whose correlation with $\hat{b}_i$ was greater than 0.9.

## 2.5 Electroanatomical mapping

Electroanatomical mapping (EAM) allows real-time reconstruction of cardiac anatomy with superimposed electrogram measures [36], using reference and mapping electrodes to determine the spatial location of the mapping catheter [37–39].

Fig 1b illustrates the data obtained from the EAM of pig 6. Sedation was administered as described in Section 2.1.2. Aligned LGE and thoracic magnetic resonance images were acquired. For thoracic imaging, visible pins were placed to identify the heart-electrode relative positions (HERP) during ECG acquisition (Fig 2a). As the hind leg electrodes were outside the field of view of the thoracic magnetic resonance, the right leg and left leg electrode coordinates were estimated by extending the right and left arm coordinates 60 cm in the antero-posterior direction.

LV endocardial mapping was performed using the Rhythmia high-resolution mapping system (Boston Scientific, Cambridge, MA, USA) [37,38]. A diagnostic catheter was percutaneously inserted into the left femoral vein and advanced into the RV to serve as an intracardiac activation reference. The Orion 64-electrode minibasket catheter was inserted into the LV through the aorta using percutaneous access made in the left carotid artery. Positional reference and impedance electrodes were placed on the right side and right back to avoid spinal interference. The mapping in sinus rhythm yielded 3939 LV endocardial points.

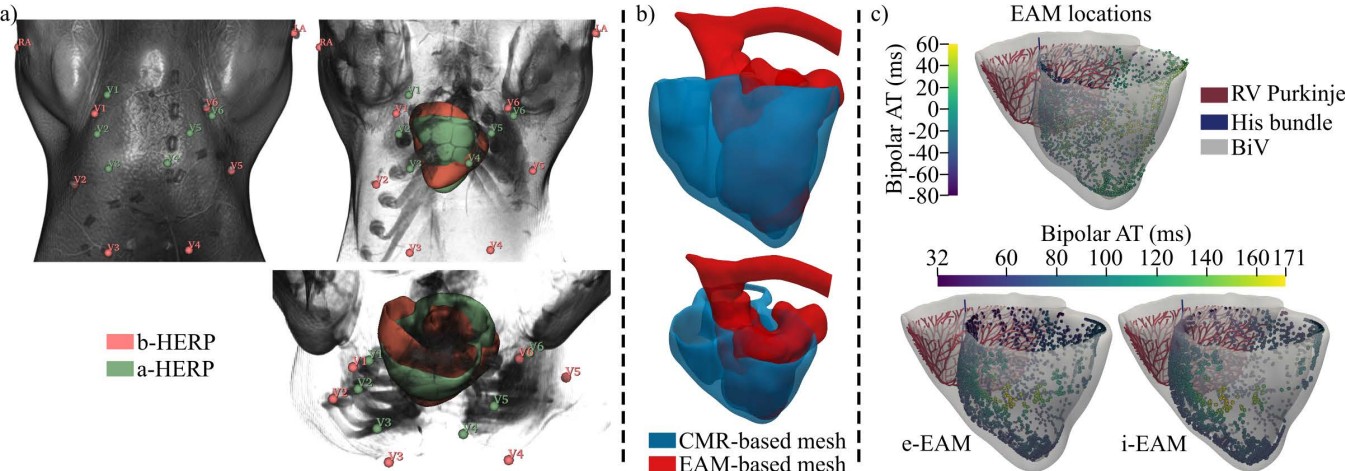

**Fig 2. Definition of HERPs and EAM-based activation.** a- Different views of the thoracic magnetic resonance of pig 6 with different opacity, showing the b-HERP (red) and a-HERP (green). RA: right arm, LA: left arm and V1-6: precordial electrodes. b- Alignment of the EAM-based mesh of the LV endocardium with the CMR-based BiV mesh. c- The BiV mesh of pig 6 is shown with the RV Purkinje, the His bundle and the preprocessed LV EAM locations with their bipolar activation (top), as well as the EAM-based endo- (e-EAM) and intramyocardial (i-EAM) bipolar activations (bottom) of the LV. b/a-HERP: Baseline/adjusted heart-electrode relative position, CMR: Cardiac magnetic resonance, AT: Activation time, LV: Left ventricle, RV: Right ventricle, BiV: Biventricular.

EAM measurements were used to define two EAM-based LV stimulation profiles for numerical simulations: endocardial EAM (e-EAM) and intramyocardial EAM (i-EAM). The aorta and the LV apex from EAM were aligned with the aortic valve and the apical LV from the CMR-based BiV mesh of pig 6, respectively (see Fig 2b). In the aligned EAM, electrograms with low bipolar voltage (≤1 mV) were excluded as SZ (or poor-contact points) [38], since SZ was defined as an insulator *in silico* (Section 2.6.3) and, therefore, its depolarization was irrelevant. Subsequently, outliers in the EAM in terms of bipolar electrogram ATs were removed, resulting in 2630 valid points (Fig 2c). Finally, the unique endocardial and intramyocardial points in the LV of the BiV model closest to each EAM site defined e-EAM and i-EAM stimulation sites. Stimulation dynamics were described by interpolating the EAM-based bipolar activation to the e-EAM or i-EAM stimulation points using radial basis functions. Thus, the BiV model was depolarized from the EAM-defined stimulation points on the LV along with the RV CS. In this context, LV depolarization was set to start 5 ms after the first RV endocardial depolarization, consistent with the mainly positive QRS complex for this pig in lead I (Fig 10a), which, based on our simulation results, suggests a more RV-to-LV activation (see discussion on the addition of Purkinje fibers to the RV septum in Section 4.2). Bipolar electrograms were used throughout, given their strong agreement with unipolar low voltage areas under high-density mapping with low inter-electrode spacing [40], as implemented here using the Orion catheter (2.5 mm spacing [38]), and their reduced susceptibility to far-field effects compared to unipolar recordings [40].

ECG signals from Rhythmia were preprocessed as described in Section 2.4. Since these ECG signals were provided in chunks that sometimes did not contain a complete beat, a full template ECG beat was selected from all recordings. Beats presenting a correlation coefficient with this template greater than 0.8 in each lead were selected, and based on them, the median ECG beat was calculated (see orange ECG in Fig 1e). Finally, the *in silico* ECG reconstructed from the two EAM-based stimulation profiles was compared with those obtained using a CS-based stimulation protocol.

## 2.6 *In silico* electrophysiological models

### 2.6.1 Anatomy.
Using the deep learning-based segmentations obtained in Section 2.2, the RV myocardium was delineated, and the final smoothed BiV surface meshes were generated for the healthy and MI cases [41,42]. Geometries

for healthy pigs were derived from end-diastolic cine-CMR, and geometries of MI pigs were derived from LGE-CMR. As shown in Fig 3b, volumetric meshes were generated via tetrahedralization with a mean edge length of 390 μm for all samples [43], consistent with previous studies [9,11,20,44].

In MI cases, the BiV domain was reconstructed using a semi-automatic approach. Initial definitions of AZ and HZ, corresponding to high and low voxel intensities, respectively, were obtained from the automatic deep learning–based segmentations of LGE images (Section 2.2.2). These LV segmentations were manually extended to the RV to generate BiV structures. Once a BiV segmentation with HZ and AZ was defined, AZ was subdivided into BZ and SZ using the full-width half-maximum method [6]. Briefly, two thresholds were defined, as shown in Fig 3a: $S_{BZ+}$, corresponding to 50% of the maximum signal intensity in AZ; and $S_{BZ-}$, corresponding to the maximum signal intensity in a remote HZ. SZ comprised AZ voxels with signal intensity greater than $S_{BZ+}$. BZ comprised AZ voxels with signal intensity in the range $[S_{BZ-}, S_{BZ+}]$. HZ, BZ, and SZ masks were interpolated onto the BiV mesh using radial basis functions.

**2.6.2 Conduction system.** Ventricular CS plays a key role in determining the electrical activation sequence that triggers mechanical contraction. CS geometry was defined from manually identified anatomical landmarks, geodesic paths, and a fractal tree algorithm.

First, the His bundle was generated by connecting landmark points in a 3D space with one-dimensional elements, following [45] (Fig 4a). Briefly, the atrioventricular node was placed 5 mm above the septal base toward the anterior RV. The His bundle extended toward the apex, bifurcating into left and right branches, both traversing from the basal to the apical endocardium along geodesic paths. The left branch was divided into the anterior and posterior left branches, while

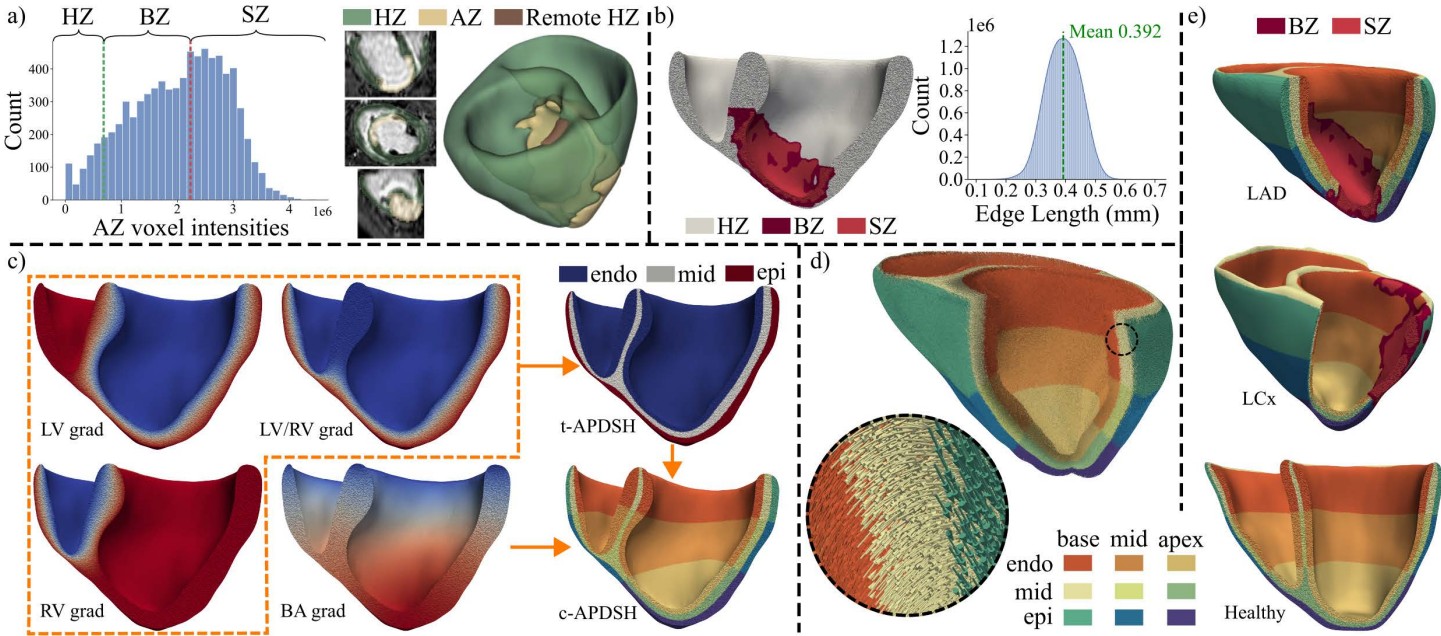

**Fig 3. *In silico* BiV healthy and MI modeling.** a- MI segmentation from LGE-CMR of the pig 10. The histogram of the AZ voxel intensities shows $S_{BZ-}$ (green) and $S_{BZ+}$ (red) thresholds delimitating the BZ. b- BZ and SZ extrapolation onto the tetrahedral mesh (left) and mesh edge length histogram for pig 10. c- BiV APDSH calculation. In the left side, the Laplace solutions representing the diffusion from LV, RV and LV/RV endocardium to epicardium (orange dashed zone) and from base to apex (BA) are shown. In the right side, the final transmural (t-APDSH) and transmural plus apicobasal (c-APDSH) BiV meshes for pig 10 are depicted. d- RBM-based BiV fiber field for Pig 10. e- Final BiV meshes for pig 10 (top), pig 6 (middle) and pig 3 (bottom). HZ: Healthy zone, BZ: Border zone, SZ: Scar zone, AZ: Affected zone, LV: Left ventricle, RV: Right ventricle, BiV: Biventricular, MI: Myocardial infarction, APDSH: Action potential duration spatial heterogeneities, LAD: Left anterior descending infarction, LCx: Left circumflex infarction, LGE-CMR: Late gadolinium-enhanced cardiac magnetic resonance, RBM: Rule-based model.

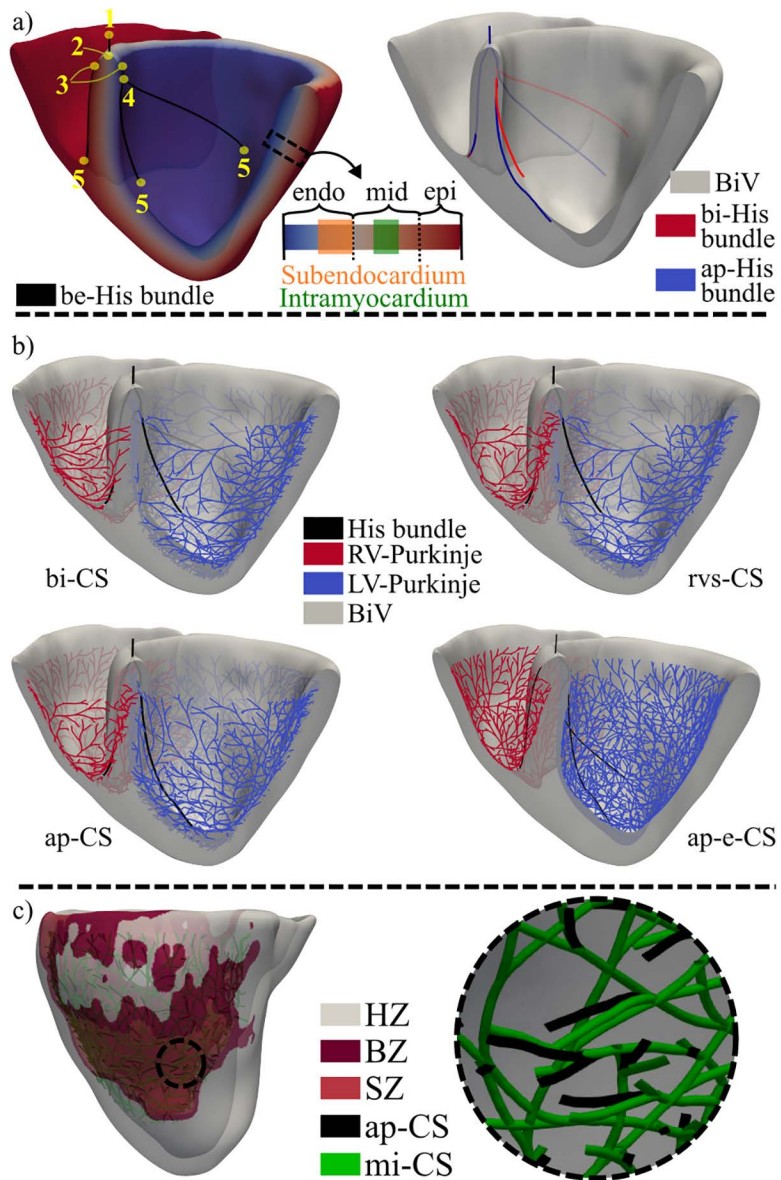

**Fig 4. Healthy (pig 3) and MI (pig 6) CS distributions.** a- In the left side, manually located landmarks for the creation of the initial be-His bundle and subendocardium and intramyocardium region definitions can be observed. The right side depicts the differences between the His bundle in the bi-CS and ap-CS. b- Coronal cut of the pig 3 BiV mesh for depiction of four CS distributions. c- Differences of the ap-CS and mi-CS zoomed in the AZ. BiV: Biventricular, MI: Myocardial infarction, CS: Conduction system, bi-CS: Baseline intramyocardial CS, rvs-CS: bi-CS with Purkinje in the septal wall of the right ventricle, ap-CS: rvs-CS with the end of the left bundle branches closer to the apex of the left ventricle, ap-e-CS: ap-CS with endocardial distribution, mi-CS: ap-CS with altered spatial distribution of Purkinje fibers following infarction, HZ: Healthy zone, BZ: Border zone, SZ: Scar zone.

the right branch remained single. Consistent with experimental studies, the LV bundle branches terminated at papillary muscles, whose insertions were identified from the end-diastolic cine-CMR [46,47], while the RV bundle branch terminated in the antero-endocardial face, more basally than the LV bundle branch endpoints and than the analogous structure in humans, in agreement with experimental observations of the septomarginal trabecula (moderator band) in pigs (see Section 4.2) [8].

The Purkinje network was generated from the endpoints of the three bundle branches using a fractal tree algorithm [48,45], resulting in an endocardial CS termed be-CS. However, given experimental evidence that the swine CS lies deeper within the myocardium than in humans [8], an alternative CS termed bi-CS was created. This bi-CS was derived by projecting be-CS into the subendocardium (the outer half of the endocardium) (Fig 4a) and, for 75% of the CS end branches in the LV and the RV, into the intramyocardial region (the middle third of the midmyocardium) [5]. PMJs were distributed throughout both ventricles, except for the basal area of the LV and the RV, and the septal and posterior regions of the RV, where activation was known to be delayed or initiated by the activation of surrounding tissue [49].

To further explore the ventricular CS architecture, four variations of the bi-CS architecture were constructed (Fig 4), and the simulation results obtained with each of them were compared with the experimental activation data:

- *rvs-CS:* Departing from bi-CS, a homogeneous distribution of PMJs was added to the RV septum.

- *ap-CS:* Departing from rvs-CS, the ends of the anterior and posterior bundle branches of the LV were positioned closer to the LV apex to accentuate the apicobasal LV depolarization.

- *ap-e-CS:* Departing from ap-CS, the entire CS was located on the endocardial surface. This served to *in silico* assess whether swine CS was better represented when projected deeper into the ventricular wall.

- *mi-CS:* Departing from ap-CS, this was modified in the AZ of MI following histological observations for LAD MI [5]. Specifically, PMJs in the AZ were relocated with a proportion of 51:36:13 for the subendocardium, intramyocardium, and epicardium, respectively. In addition, the number of PMJs was reduced to 42% of the total amount in each of the three layers (see Fig 4c).

**2.6.3 Electrophysiology.** APs of healthy porcine ventricular myocytes were described using the biophysically detailed model by Gaur et al. [10], while the APs of CS cells were represented with the Stewart et al. [50] model. All ventricular regions were assigned orthotropic conductivity with transverse isotropy. The longitudinal diffusion coefficient (LDC) values were 0.0013 cm$^2$/ms for HZ [51] and 0.013 cm$^2$/ms for CS (CV ≈ 200 cm/s [49]), which were respectively equivalent to 0.13 and 1.3 S/m for a membrane surface-to-volume ratio of 1000 cm$^{-1}$ and capacitance per unit area of 1 µF/cm$^2$. These values defined the basal LDC (b-LDC) setting. CS LDC was reduced near its end points following a sigmoid curve to smoothly match the ventricular LDC [52]. The transverse-to-longitudinal diffusion ratio was set to 0.25 for HZ, in agreement with the transverse anisotropy reported for cardiac CV [53].

For MI BZ, excitability was reduced by decreasing $g_{Na}$ to 38% of the HZ value, following [11,20,54]. Conductivity and APD were then adjusted individually for each LAD-MI pig by iteratively modifying LDC and the conductance of the inward rectifier potassium current ($g_{K1}$) until the simulated CV and APD$_{90}$ matched OM measurements. As detailed in Table 1, the individualization of these parameters was not feasible for all pigs due to the absence of the necessary experimental electrophysiological data. In these specific cases, the LDC and $g_{K1}$ parameters were instead assigned the mean values calculated from the subset of pigs for which experimental data were available (see, for example, LDC definition in Table 4).

SZ was modeled as an insulator [11,20], and thus a zero-flux Neumann condition was imposed at its boundary. Furthermore, from the comparison of the median fractional anisotropy values in the healthy and MI regions of the DW-CMR data, a mean decrease of 38% in anisotropy values was found in the MI region with respect to the healthy region. Specifically, the fractional anisotropy decreased from 0.31 in HZ to 0.18 in AZ (41%) in pig 7 and from 0.63 to 0.41 (34%) in pig 14. Consequently, the transverse-to-longitudinal diffusion ratio was increased from 0.25 to 0.345 (38%) in BZ.

**2.6.4 Fiber orientation.** When DW-CMR data are unavailable, rule-based models (RBMs) are often used to define the cardiac fiber field in *in silico* ventricular models [55]. RBMs establish a local coordinate system composed of circumferential, apicobasal, and transmural directions and define the rotation angles *α* and *β* for the RV, LV, and septal endocardium and epicardium. The angle *α* represents the rotation with respect to the clockwise circumferential direction calculated as if the heart was seen in the apicobasal direction. The angle *β* describes the rotation of the outward

transmural direction. Based on the values of these angles, the longitudinal, sheet, and sheet-normal cardiomyocyte directions are defined for all regions of a BiV model. Here, DW-CMR data were combined with an RBM to generate porcine-specific BiV fiber fields.

The RBM of Bayer et al. [55] was used, with septal properties matched to the LV and all $\beta$ angles set to 0° (transverse isotropy). Each sample thus required four parameters: $\alpha$ angles in the endocardium ($\alpha_{endo}$) and epicardium ($\alpha_{epi}$) for both LV and RV. A standard fiber field (s-RBM) was defined using $\alpha$ values of +60° and -60° for the endocardium and epicardium, respectively.

For the three healthy pigs with DW-CMR data (Table 1), RBM-generated fiber fields (with varied $\alpha$ angles) were compared to DW-based fiber fields. Because ex vivo DW-based geometries differed from in vivo cine-based geometries, it was difficult to extrapolate the fiber field; thus, DW-based RBMs (dw-RBMs) were defined for cine-based anatomical models of pigs 1–3. For this, ex vivo BiV tetrahedral meshes were created from the segmentation of the DW-CMR data of each sample, with the meshes having a mean edge length of 1 mm, which matches the voxel resolution of the DW-CMRs (Table 2). The DW-based fiber field was then interpolated using radial basis functions. RBMs were calculated for each ex vivo BiV mesh by initially setting the angles $\alpha$ in the LV to the default values of [+60°, -60°] while the RV angles $\alpha_{endo}$ and $\alpha_{epi}$ varied in the range [+90°, -90°) in 10° steps, producing 324 RBM-based fiber fields per mesh. These were compared with the DW-based fiber field to identify the pair of RV angle values [$\alpha_{endo}$, $\alpha_{epi}$] minimizing the mean angle $\overline{\Theta}$, defined from the longitudinal fiber directions as [55]:

$$\overline{\Theta} = \frac{1}{M} \sum_{m=0}^{M} cos^{-1} \left( \frac{\boldsymbol{u}_m \cdot \boldsymbol{v}_m}{||\boldsymbol{u}_m|| \, ||\boldsymbol{v}_m||} \right)$$

where $M$ is the total number of nodes and $\boldsymbol{u}_m$ and $\boldsymbol{v}_m$ are the longitudinal fiber directions in node $m$ obtained from RBM and DW-CMR, respectively. RV $\alpha$ angles were further refined by varying them around the identified values, specifically in the range [$\alpha_x - 10, \alpha_x + 10$] in steps of 1°, with $x$ being endo or epi. The values of the angles associated with the lowest $\overline{\Theta}$ were selected. To complete the determination of RBM-based fiber fields, the same procedure was repeated by setting the RV $\alpha$ angles to their optimal values and varying the $\alpha$ angles in the LV to identify their optimal values.

**2.6.5 Repolarization heterogeneities.** Three APD spatial heterogeneity (APDSH) configurations were defined (Fig 3c): b-APDSH, representing homogeneous APD throughout the BiV mesh; t-APDSH, including transmural APD heterogeneities; and c-APDSH, including transmural and apicobasal APD heterogeneities.

To include transmural heterogeneities, the steady-state diffusion (Laplace) equation was solved multiple times with different boundary conditions [56] to accurately segment the BiV mesh transmurally (orange dashed region in Fig 3c). A proportion of 40:35:25 was defined for the endocardium, midmyocardium, and epicardium, following previous studies [57,58]. To include apicobasal heterogeneities, a base-to-apex gradient was defined by solving the Laplace equation with constant Dirichlet boundary conditions at the apex and base of the BiV domain. Based on this gradient and to divide the BiV mesh into three apicobasal regions, a proportion of 48:35:17 was empirically established for the base, middle, and apex regions (Fig 3c).

The Gaur et al. [10] cellular AP model lacks native repolarization gradients, so experimental porcine data from Meijborg et al. [59] were used to define APD heterogeneities. The APD of the Gaur et al. [10] model was taken to correspond to the midmyocardium and the middle height between the apex and the base [10;60]. Since APD in the model is largely determined by the expression of $I_{K1}$, as reported by the authors, APD in each of the nine different regions (3 transmural per 3 apicobasal) was defined by varying the conductance $g_{K1}$ to match the target APD values within 0.5 ms (Table 3). The nine target $APD_{90}$ values were set by varying the default $APD_{90}$ value in the Gaur model, measured at a cycle length of 650 ms, according to the percentage of variations in repolarization reported in [59]. For b-APDSH, the midmyocardial middle-height $g_{K1}$ value was used. For t-APDSH, the endocardial, midmyocardial, and epicardial middle-height $g_{K1}$ values were used.

**Table 3. Definition of repolarization heterogeneities.** The target (T) and simulated (S) $APD_{90}$ values (at 650 ms cycle length) are shown together with the $g_{K1}$ (x10) values leading to the corresponding simulated $APD_{90}$. B: Base, M: Middle, A: Apex, En: Endo, Ep: Epi, $g_{K1}$: Maximum conductance of the inward rectifier K+ current.

| | B | | | M | | | A | | |
|---|---|---|---|---|---|---|---|---|---|
| | T (ms) | S (ms) | $g_{K1}$ (mS/μF) | T (ms) | S (ms) | $g_{K1}$ (mS/μF) | T (ms) | S (ms) | $g_{K1}$ (mS/μF) |
| En | 213.9 | 213.9 | 1.49 | 222.7 | 222.4 | 1.39 | 218.3 | 218.1 | 1.44 |
| M | 204.8 | 204.5 | 1.61 | 213.2 | 213.2 | 1.5 | 208.9 | 208.5 | 1.56 |
| Ep | 199.0 | 199.4 | 1.68 | 207.2 | 207.3 | 1.57 | 203.1 | 203.1 | 1.63 |

## 2.7 Numerical simulations

Seven simulation groups (G1 to G7) were defined, each with specific CS, fiber orientation, repolarization heterogeneities, tissue conductivities, and HERP configuration (Table 4). Even though complete experimental datasets were not available for every pig, we carefully designed our modeling framework to maximize individualization whenever possible, both at the

**Table 4. Simulation groups.** Parameters in square brackets were varied in the simulation group for evaluation. Average LDC ($\overline{LDC}$) and $g_{K1}$ ($\overline{g_{K1}}$) values were defined as the mean of the OM-adjusted $LDC^i$ and $g_{K1}^i$ values at the BZ of pigs 8-12, therefore, i=8,...,12. PMJ: Purkinje-muscular junction, PMJr: PMJ radius, b-LDC: Baseline longitudinal diffusion coefficient, $g_{K1}$: Maximum conductance of the inward rectifier K+ current, OM: Optical mapping, MR: Magnetic resonance, LGE: Late gadolinium-enhanced, DW: Diffusion-weighted, T: Thoracic, CS: Conduction system, bi-CS: Baseline intramyocardial CS, rvs-CS: bi-CS with Purkinje in the septal wall of the right ventricle, ap-CS: rvs-CS with the end of the left bundle branches closer to the apex of the left ventricle, ap-e-CS: ap-CS with endocardial distribution, mi-CS: ap-CS with altered spatial distribution of Purkinje fibers following infarction, e/i-EAM: Endocardial/intramyocardial activation based on the electroanatomical mapping, s/dw-RBM: Standard/diffusion-weighted rule-based model, b/t/c-APDSH: Baseline/transmural/transmural-apicobasal action potential duration spatial heterogeneities, HZ: Healthy zone, BZ: Border zone, b/a-HERP: Baseline/adjusted heart-electrode relative position, ECG: Electrocardiogram.

| Group (pig) | Input (pig) | Activation | PMJr (mm) | RBM | APDSH | LDC | HERP | Validation (pig) |
|---|---|---|---|---|---|---|---|---|
| G1 (1) | cine-MR (1) | [be-CS, bi-CS] | [0.1-2] | s- | b- | b- | – | – |
| G2 (1–3) | cine-MR (1–3) | [bi-CS, rvs-CS, | 0.5 | s- | b- | b- | [b-, a-] | ECG (15) |
| | T-MR (6) | ap-CS, ap-e-CS] | | | | | | |
| G3 (1–3) | cine-MR (1–3) | ap-CS | 0.5 | s- | [b-, t-, c-] | b- | a- | ECG (15) |
| | T-MR (6) | | | | | | | |
| G4 (1–3) | cine-MR (1–3) | ap-CS | 0.5 | [s-, dw-] | c- | b- | a- | ECG (15) OM (13) |
| | DW-MR (1–3) | | | | | | | |
| | T-MR (6) | | | | | | | |
| G5 (8–12) | LGE-MR (8–12) | ap-CS | 0.5 | s- | HZ: c- BZ: $g_{K1}^i$ | HZ: b- BZ: $LDC^i$ | – | OM (8–12) |
| | OM (8–12) | | | | | | | |
| | DW-MR (7,14) | | | | | | | |
| G6 (6) | LGE-MR (6) | [ap-CS, mi-CS, e-EAM, i-EAM] | 0.5 | s- | HZ: c- BZ: $\overline{g_{K1}}$ | HZ: b- BZ: $\overline{LDC}$ | b- | ECG (6) |
| | EAM (6) | | | | | | | |
| | DW-MR (7,14) | | | | | | | |
| | OM (8–12) | | | | | | | |
| | T-MR (6) | | | | | | | |
| G7 (4–12) | LGE-MR (4–12) | ap-CS | 0.5 | s- | HZ: c- BZ: $\overline{g_{K1}}$ | HZ: b- BZ: $\overline{LDC}$ | b- | – |
| | DW-MR (7,14) | | | | | | | |
| | OM (8–12) | | | | | | | |
| | T-MR (6) | | | | | | | |

species-specific level and, when feasible, at the subject-specific level. As shown in Table 4, experimental data were used for input (feeding) and validation of the modeling and simulation:

- *CMRs: In vivo* data were used to individually define the geometry of the 12 BiV models generated: the end-diastolic image from cine-CMR for the 3 healthy cases and the LGE-CMR for the 9 MI (4 LCx and 5 LAD) cases. DW-CMR data were used for subject-specific calibration of the rule-based fiber orientation in healthy cases (DW-CMR from pigs 1–3 in G4) and to define species-specific reductions in electrical conductivity anisotropy in MI cases (DW from pigs 7 and 14 in G5-7).

- *Thoracic magnetic resonance:* Images of pig 6 were used to the define the HERP for pigs in simulation groups assessing the ECG (G2-4 and G6-7).

- *OM:* Mapping data were used exclusively for validation in the G4 simulations, whereas in the G5 simulations they were used for subject-specific model calibration and validation. In G4, *in silico* ATs, depolarization patterns, CVs, and $APD_{90}$ values were validated against their OM-obtained counterparts. In G5, OM measurements of CV and $APD_{90}$ in the BZ were used to calibrate exclusively the values of LDC and $g_{K1}$ in BZ for MI pigs 8–12, while ATs, depolarization patterns, and the values of CV and $APD_{90}$ in the HZ were used for validation. The resulting subject-specific calibration in G5 was used in the following G6-7 simulations for defining average species-specific values of LDC and $g_{K1}$ in BZ for the MI models.

- *EAM:* Mapping data from pig 6 were used to define a subject-specific post-MI activation of the LV in pig 6 at the G6 simulations.

- *ECGs:* The healthy ECG recording from pig 15 was used to validate species-specific CS distributions and repolarization heterogeneities (G2-3), while the MI ECG recording was used to assess ECG replication under post-MI conditions, including species-specific alterations on PMJ spatial distributions or the incorporation of subject-specific activation pattern into the MI model of pig 6 (G6).

Cellular models were prepaced for at least 2000 beats until the $APD_{90}$ reached steady state (i.e., remained invariant). Tissue propagation was computed by numerically integrating the monodomain model [61]. Each BiV simulation consisted of 3 cardiac cycles, with the last cycle being used for analysis. Stimuli of 80 μA/cm$^2$ magnitude and 1 ms duration were applied at the atrioventricular node. $V_m$ was saved with a time step of 0.25 ms and the adaptive integration time step $\Delta t$ was searched in the range [0.01,0.125] ms [61]. The cycle duration for the cell and tissue simulations depended on validation data: 769 ms for comparisons with pig 15 ECG (G2-4), 898 ms for comparisons with pig 6 ECG (G6), and 1000 ms for comparisons with OM data (G4-5) or without comparison with experiments (G1 and G7).

A total of 74 simulations across all pigs was performed using the open source software ELECTRA (v0.6.2) on the HERMES HPC cluster at the University of Zaragoza.

## 2.8 Postprocessing

In G1 simulations, the CS-myocardium coupling was analyzed by varying the PMJ radius for the two types of CS, i.e., be-CS and bi-CS. Three metrics were calculated: the percentage of PMJs with at least one CS-to-myocardium line element; the mean number of connections from the CS to the myocardium per PMJ; the percentage of PMJs where at least one tissue node had an AT difference with the penultimate CS node for the corresponding PMJ in the range [0, 1.5] ms, which implied PMJ activation from the CS to the myocardium (that is, no retrograde activation).

In G1, G4, and G5 simulations, AT, $APD_{90}$, and CV were calculated. AT was the time from a fixed time reference to the time when the AP upstroke crossed 0 mV. $APD_{90}$ was the time difference between the time point with the maximum voltage derivative in the AP upstroke and the time corresponding to 90% repolarization from the AP peak to the diastolic $V_m$. CV was calculated for each node of the BiV mesh. The magnitude of the velocity vector between two nodes was

calculated as the ratio of the Euclidean distance between them and the difference in their ATs. For each node, the mean CV vector between the given node (considered as the central node) and all its neighboring nodes within a radius of 600 μm was determined. The radius value was chosen to cover the maximum edge length found in all meshes (Fig 3b for an example). For G4 and G5 simulations, the anterior (and posterior for G4) surface of the BiV mesh was cut after retrieving the simulation results, and CV and $APD_{90}$ were compared with two-dimensional (2D) OM data.

Pseudo-ECGs (pECGs) were defined from extracellular unipolar potentials $\phi(\boldsymbol{e})$, which were approximated by the integral of the dipole source density as in [21]:

$$\phi(\boldsymbol{e}) = \int_{\Omega} \left( -\mathbf{D}\nabla V_m \cdot \left( \nabla \frac{1}{\|\boldsymbol{r} - \boldsymbol{e}\|} \right) \right) \, d\boldsymbol{r}$$

where $\boldsymbol{e}$ is the vectorial electrode position, $\mathbf{D}$ is the diffusion tensor, and $\Omega$ is the BiV domain. Once the 12-lead pECG was drawn from the calculated $\phi(\boldsymbol{e})$ signals, a low-pass Butterworth filter with a 40 Hz cutoff frequency was applied, as for experimental ECGs in Sections 2.4 and 2.5. Then, the pECG and the experimental ECG signals were normalized in each of the leads and their baseline was set to zero [62]. Note that the 12-lead pECG was drawn from the third simulated beat (Section 2.7) and the 12-lead experimental ECG corresponded to the median beat obtained as described in Sections 2.4 and 2.5.

For the comparison of simulations and experiments, we separately aligned the QRS complexes and the T waves by segmenting them. Specifically, four mean peak times were calculated corresponding to the QRS complex and the T wave in the simulated and experimental data sets. Each mean peak time was computed as the average of the peak times on the 12 leads for the wave of interest (QRS complex or T wave). The signals were segmented in each lead by taking [-50,50] ms for the QRS complex and [-100,100] ms for the T wave, centered around the corresponding mean peak time. Pearson correlation coefficients were calculated for the QRS complexes and T waves in each lead to quantify the similarity between simulated and experimental waveforms. These coefficients served to provide a normalized metric of shape similarity rather than a basis for any statistical inference.

In another group of simulations, the impact on the calculation of pECGs of the heart position and the location of the electrodes in the torso was evaluated. Since pig 6 was the only one in which the exact HERP was known, for subsequent pECG computations, all BiV models of pigs 1–12, except for pig 6, were spatially matched to the BiV mesh of pig 6 using the iterative closest point algorithm, which defined the baseline HERP (b-HERP) for each BiV model. To perform this rigid transformation, reference points were manually located in the anterior, posterior, left, right, and septal regions of the base, as well as in the endocardial and epicardial apex of the LV in all meshes. In G2-4 simulations, the experimental ECG used for validation was measured in a different swine (pig 15) with a likely different HERP due to a distinct spatial location of the heart and / or unequal positioning of the ECG electrodes. Thus, not only were several CS distributions tested with b-HERP, but also rotations of the heart position and variations in the electrode location for precordial lead measurement (so-called adjusted HERP, a-HERP) were evaluated in G2. The optimal CS distribution and the optimal a-HERP that, in combination, minimized the mismatch between the pECGs of pigs 1–3 and the experimental ECG of pig 15 were identified. In G3-4 and G6-7, a-HERP and b-HERP were used for the calculation of pECG, respectively. In G6, b-HERP was the exact HERP, and in G7, pECGs for different infarcted porcine ventricles were compared (Table 4).

## 3 Results

### 3.1 Optimization of PMJ radius for effective CS-myocardium coupling

In G1 simulations, multiple PMJ radius values for both be-CS and bi-CS were tested, and their effects on the numerical activation of the BiV model of pig 1 were evaluated. Table 5 and Fig 5 summarize the results. At small PMJ radii as 0.1 mm, the percentage of connected PMJs was low (28.5% for be-CS, 10.4% for bi-CS), resulting in limited number of

**Table 5. CS-myocardium coupling characteristics.** be/bi-CS: Baseline endocardial/intramyocardial conduction system, PMJ: Purkinje-muscular junction, PMJr: PMJ radius, ncPMJ: percentage of connected PMJs, cpPMJ: mean number of connections per PMJ, aPMJ: percentage of PMJs with anterograde activation.

| CS | PMJr (mm) | ncPMJ [%] | cpPMJ | aPMJ [%] |
|---|---|---|---|---|
| be- | 0.1 | 28.48 | 1.00 | 28.13 |
| be- | 0.5 | 100 | 11.97 | 93.78 |
| be- | 1.0 | 100 | 76.83 | 90.01 |
| be- | 1.5 | 100 | 240.51 | 23.05 |
| be- | 2.0 | 100 | 550.59 | 1.05 |
| bi- | 0.1 | 10.43 | 1.00 | 6.92 |
| bi- | 0.3 | 100 | 3.21 | 90.27 |
| bi- | 0.5 | 100 | 14.51 | 93.34 |
| bi- | 0.6 | 100 | 25.21 | 93.69 |
| bi- | 0.9 | 100 | 84.91 | 37.07 |
| bi- | 1.2 | 100 | 201.24 | 1.23 |

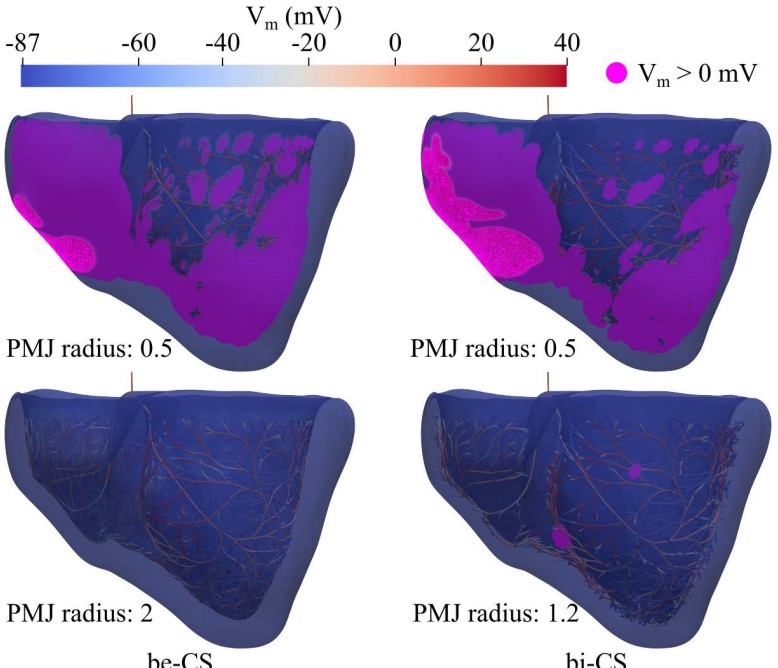

**Fig 5. Effect of PMJ radius on CS-myocardium coupling.** Simulated CS-myocardium depolarization for pig 1 when be-CS and bi-CS with different PMJ radius values (mm) were used. Snapshots after 45 ms of atrioventricular node stimulation are shown. Myocardial nodes with $V_m \geq 0$ mV are represented in purple. PMJ: Purkinje-muscular junction, $V_m$: Transmembrane potential, be/bi-CS: Baseline endocardial/intramyocardial conduction system.

PMJs with anterograde activation (28% for be-CS, 7% for bi-CS). Increasing the radius to 0.5 mm for be-CS and 0.3 mm for bi-CS ensured complete connection of Purkinje endpoints to the myocardium (100% of PMJs were connected with the myocardium) and markedly increased the number of PMJs with anterograde activation (93.8% for be-CS, 90.3% for bi-CS).

The maximum percentages of PMJs with anterograde activation occurred at a PMJ radius of 0.5 mm for be-CS and 0.6 mm for bi-CS, with 12 and 25 mean number of connections per PMJ, respectively. Further increases in PMJ radius

increased the mean number of connections per PMJ but reduced the number of PMJs with anterograde activation. For example, the percentage of PMJs with anterograde activation decreased from approximately 90% to 23% (be-CS, radius 1 to 1.5 mm) and to 37% (bi-CS, radius 0.6 to 0.9 mm). At large radii, the percentage of PMJs with anterograde activation became negligible (1%), with the mean number of connections per PMJ exceeding 200 connections.

Given that all BiV models had the same mesh resolution, a PMJ radius of 0.5 mm was selected for G2-7 simulations to avoid conduction block at PMJs.

### 3.2 Ventricular activation and QRS morphology shaped by CS distribution and electrode position

In G2 simulations, the experimentally recorded QRS of pig 15 was compared with the simulated QRS of pigs 1–3 to evaluate how different CS and electrode positions (defined by HERP configurations) influence ventricular activation.

*Effect of CS distribution with fixed HERP configuration*: When comparing bi-CS and rvs-CS for both b-HERP and a-HERP, rvs-CS consistently improved agreement with experimental QRS morphology, particularly in leads I, II, aVR, and aVL (Fig 6a, pig 3). For b-HERP, these effects were even more pronounced, with mean QRS similarity across the three pigs increasing from -0.51, 0.31, -0.20 and -0.84 for bi-CS to 0.96, 0.91, 0.97 and 0.91 for rvs-CS in leads I, II, aVR, and aVL, respectively.

*Effect of HERP configuration with fixed CS distribution*: For bi-CS, switching from b-HERP to a-HERP improved QRS similarity in precordial leads V2-V5 from -0.8, -0.61, -0.3 and -0.25 to 0.2, 0.7, 0.8, and 0.84. For rvs-CS, the change from b-HERP to a-HERP yielded improvements from -0.01, -0.96, -0.93, and -0.93 to 0.86, 0.91, 0.73, and 0.48 (Fig 6a). However, in leads V1 and V6, a-HERP reduced QRS similarity when used with bi-CS, whereas similarity remained high with rvs-CS. Despite improved precordial replication with rvs-CS combined with a-HERP, limb and augmented lead similarity (especially lead III) decreased, as shown in Fig 6a.

*Effect of ap-CS*: With HERP fixed to a-HERP, ap-CS outperformed bi-CS in all leads except aVF (Fig 6c). In particular, QRS similarity in lead III increased to 0.56, while it decreased to -0.15 in lead aVF.

*Overall effects*: Mean QRS similarity across all samples and leads was -0.3, 0.1, and 0.23 with b-HERP and -0.16, 0.5, and 0.75 with a-HERP for bi-CS, rvs-CS, and ap-CS, respectively. Thus, QRS replication improved progressively with rvs-CS and ap-CS, particularly when combined with a-HERP (Fig 6b).

*Effect of endocardial vs intramyocardial activation*: Comparison of ap-e-CS with ap-CS under a-HERP revealed a reduction in QRS similarity in all leads when activation was restricted to the endocardium (Fig 6d). Mean QRS similarity dropped from 0.75 to 0.35, with the largest decreases in leads III (54.3%), aVL (42.8%), V2 (28%), and V6 (31.4%).

### 3.3 Ventricular repolarization heterogeneities and T-wave variations

In G3 simulations, the influence of transmural and apicobasal APD heterogeneities on T-wave reproduction was examined.

The experimental T waves of pig 15 were compared qualitatively (Fig 7a) and quantitatively (Fig 7b) with simulations for pigs 1–3 using b-APDSH, t-APDSH, and c-APDSH. Both t-APDSH and c-APDSH improved T-wave similarity in all leads except I and aVR. The greatest improvements occurred in leads II, III, aVL, and aVF, where mean similarity rose from 0.1, -0.25, 0.62, and -0.57 (b-APDSH) to 0.8, 0.91, 0.89, and 0.92 (c-APDSH). In leads I and aVR, mean similarity decreased by 17% and 29%, although for pig 1, c-APDSH still produced visually similar T waves to the experiment in these leads (Fig 7a). Overall, incorporating heterogeneous APDs (t-APDSH or c-APDSH) yielded more biomimetic T waves for all pigs and precordial leads, although similarity in lead V6 was consistently low.

In terms of the QRS complex, APD heterogeneities had a negligible effect, with mean similarity values of 0.756, 0.753, and 0.756 for b-APDSH, t-APDSH, and c-APDSH, respectively. In contrast, T-wave similarity increased markedly, from 0.49 (b-APDSH) to 0.75 (t-APDSH) and 0.74 (c-APDSH). Moreover, good agreement between the experimental and simulated duration of the ST segment was achieved. This agreement can be observed in

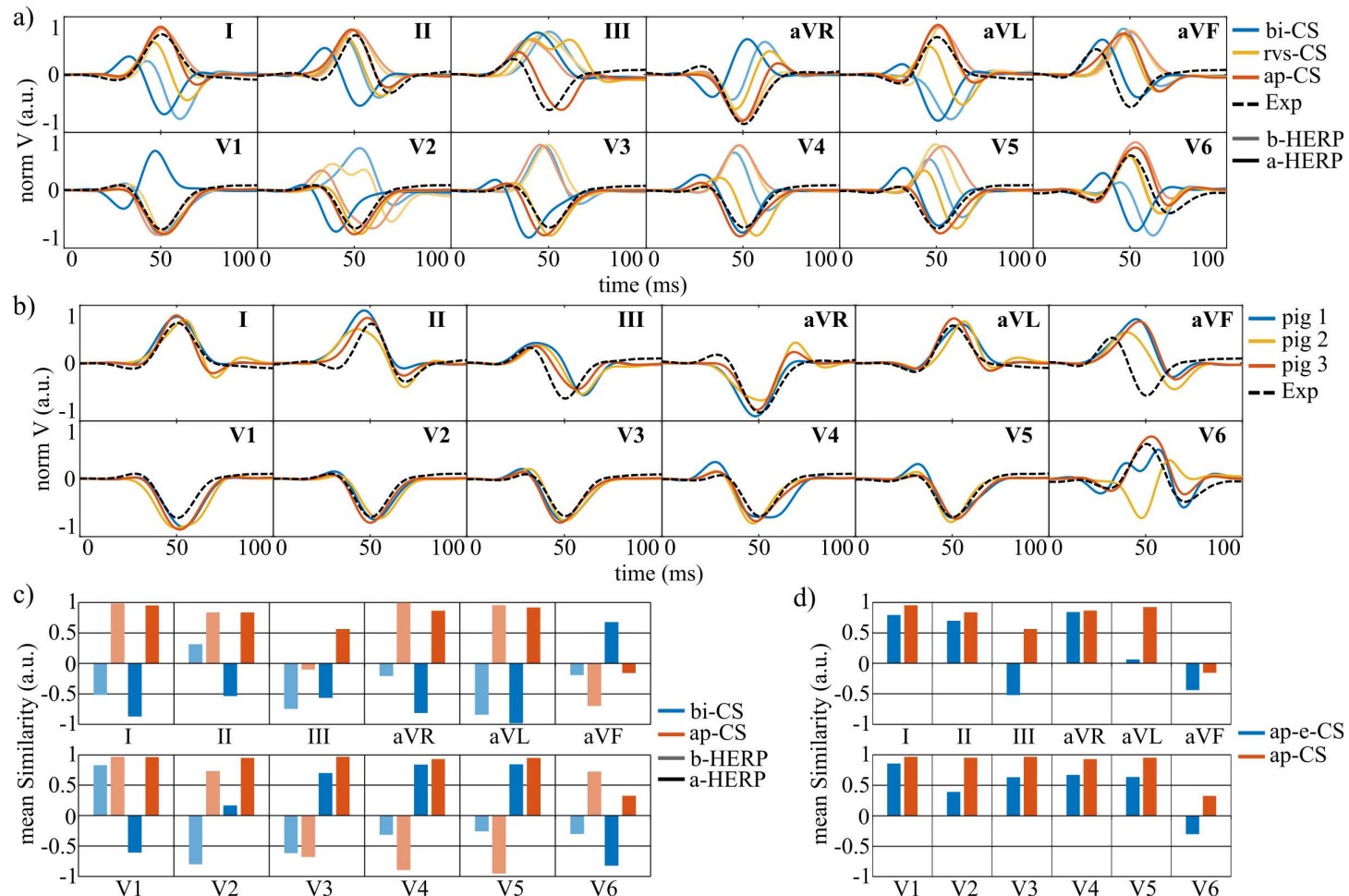

**Fig 6. Effect of CS distribution and HERP on simulated QRS complexes.** a- Simulated QRS complexes for pig 3 when different HERP and CS distributions were used are depicted with the experimental QRS complexes from pig 15 (Exp, black dashed line). Semi-transparent and solid lines represent the results when the b-HERP and a-HERP were used, respectively. b- Simulated QRS complexes obtained with a-HERP and ap-CS for pigs 1-3 are depicted with the experimental QRS complexes from pig 15 (Exp, black dashed line). c- Simulated versus experimental mean QRS similarity obtained for pigs 1-3 when different HERP and CS distributions were employed. d- Mean QRS similarity for pigs 1-3 for simulations with ap-e-CS and ap-CS distributions. CS: Conduction system, bi-CS: Baseline intramyocardial CS, rvs-CS: bi-CS with Purkinje in the septal wall of the right ventricle, ap-CS: rvs-CS with the end of the left bundle branches closer to the apex of the left ventricle, ap-e-CS: ap-CS with endocardial distribution, b/a-HERP: Baseline/adjusted heart-electrode relative position.

S2 Fig, which shows the complete simulated and experimental ECG waveforms for healthy pigs 1–3 and pig 15, respectively.

## 3.4 Electrophysiological effect of subject-specific myocardial fiber orientation calibration

In G4 simulations, myocardial fiber orientations obtained by individually adjusting the RBM to each pig's DW-CMR data (pigs 1–3, Section 2.6.4) were compared with those generated by s-RBM. The greatest deviation from s-RBM was observed for pig 1. After DW-CMR-based adjustment, LV $\alpha$ angles were -100° (endocardium) and -91° (epicardium), while RV $\alpha$ angles were 86° and 73°, respectively. The smallest difference occurred in pig 2, whose DW-RBM produced LV $\alpha$ angles of 53° and -62° and RV $\alpha$ angles of 64° and -51°, closely matching s-RBM.

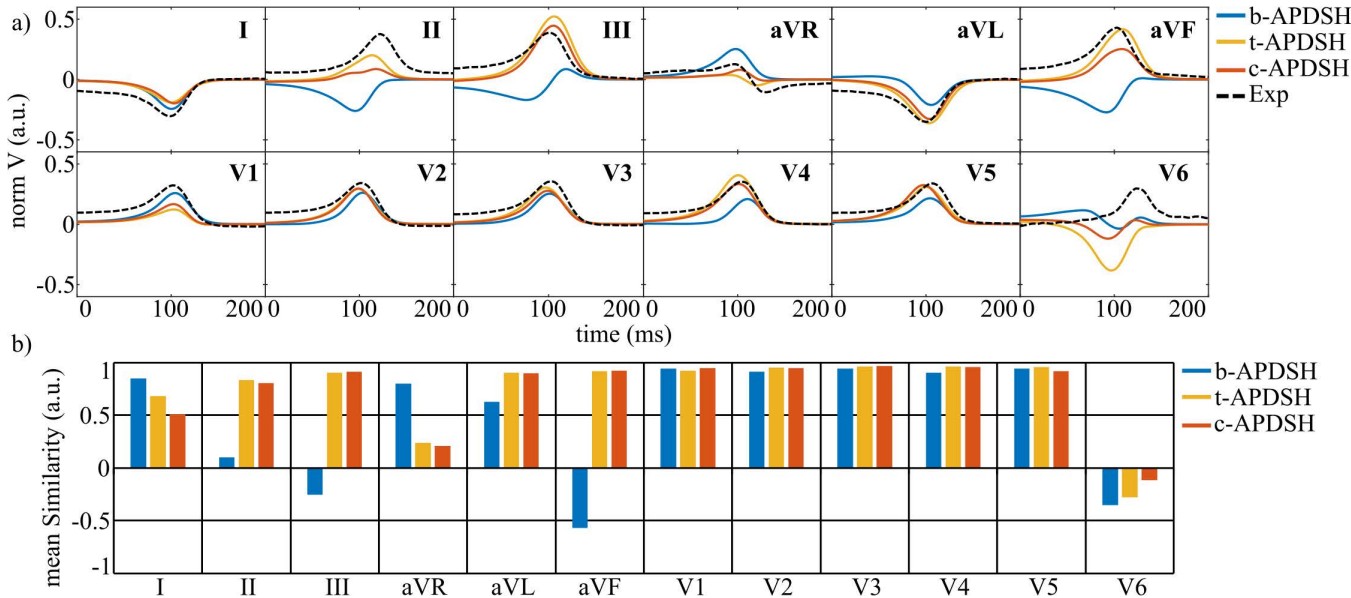

**Fig 7. Effect of APDSH on simulated T waves.** a- Simulated T waves for pig 1 when different APDSH were used. b- T-wave mean similarity for simulations of pigs 1-3 for b-APDSH, t-APDSH, and c-APDSH. b/t/c-APDSH: Baseline/transmural/transmural-apicobasal action potential duration spatial heterogeneities, Exp: the experimental T wave from pig 15 is depicted as a black dashed line.

At the macroscopic level, pECGs were largely unaffected by replacing s-RBM with dw-RBM. Fig 8a compares pECGs for pigs 1 and 2, representing the lowest and highest similarity between s-RBM and dw-RBM, respectively. Across all simulations, the mean QRS similarity between the experimental and simulated signals decreased slightly from 0.76 to 0.73, while T-wave similarity increased from 0.74 to 0.76 when using dw-RBM.

At the tissue level, activation patterns were nearly identical between RBMs (Fig 8b, right, for pig 1). Almost all differences in maximum AT were below 1 ms. Only a discrepancy of 8.5 ms in maximum AT was observed in the posterior view of pig 1.

Comparisons with the OM-based AT map for pig 13 revealed similar anterior and posterior AT and activation patterns for both RBMs (Fig 8b). In both OM and simulations, epicardial activation began in the RV, propagated toward the LV, and concluded at the LV base. Maximum AT values were comparable between OM (46/50 ms for anterior/posterior) and simulations (in pig 1, s-RBM yielded 43/36 ms and dw-RBM 44/44.5 ms).

CV analysis showed a modest 3.6 cm/s reduction in median CV with dw-RBM relative to s-RBM (Fig 8c), while median $APD_{90}$ remained identical at 246 ms in both anterior and posterior views. Relative to OM experiments for pig 13 and HZ of MI cases, median $APD_{90}$ differed by 5.4%, whereas simulated median CV exceeded experimental values by 31% (s-RBM) and 26% (dw-RBM). Despite these offsets, CV distributions in the anterior view were similar between dw-RBM simulations and OM (Fig 8d). Averaging dw-RBM results for pigs 1–3 showed anterior CVs only 8% above OM values, while posterior CVs remained approximately 30% higher (experimental medians: 84.3 / 71.7 cm/s for anterior / posterior).

### 3.5 Lower cellular coupling and inward rectifier potassium mimic MI electrical dynamics

OM data from the anterior view of MI pigs with LAD occlusion (pigs 8–12) were used to calibrate the *in silico* MI BZ electrophysiology in G5 simulations. Table 6 lists the LDC and $g_{K1}$ values yielding optimal agreement in CV and $APD_{90}$. The relative error between the experimental and simulated median CV of pigs 8–12 was 3.2%, 3.9%, 1.0%, 2.9%, and 0.2%

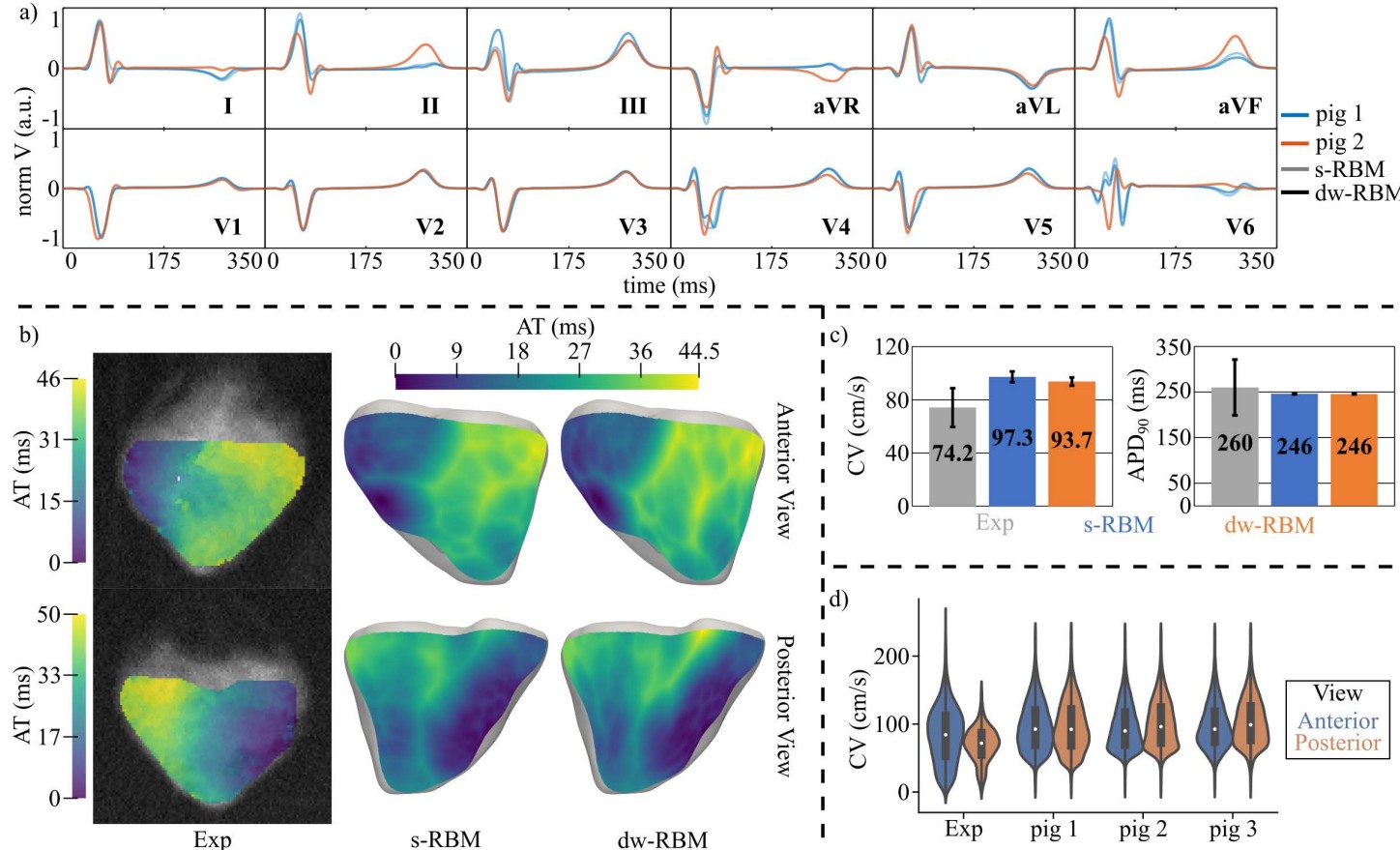

**Fig 8. Effect of cardiomyocyte orientation.** a- *In silico* ECGs for pigs 1-2 simulated using s-RBM and dw-RBM. b- Experimental (left) and simulated (right) AT maps for different cardiac views. The experimental maps correspond to pig 13 and the simulated maps to pig 1 using s-RBM and dw-RBM. c- Mean and standard deviation of CV and APD$_{90}$ values for all OM experiments and simulations of pigs 1-3 with s-RBM and dw-RBM. d- CV distribution for the anterior and posterior views of the OM data of pig 13 and the simulations of pigs 1-3 when dw-RBM was used. s/dw-RBM: Standard/ diffusion-weighted rule-based model, AT: Activation time, CV: Conduction velocity, APD$_{90}$: Action potential duration at 90% repolarization.

**Table 6. OM-based CV and APD$_{90}$ tuning for the BZ of MI.** OM: Optical mapping, CV: Conduction velocity, APD$_{90}$: Action potential duration at 90% repolarization, BZ: Border zone, MI: Myocardial infarction, LDC: Longitudinal diffusion coefficient, $g_{K1}$: Maximum conductance of the inward rectifier K$^+$ current, E: Experiment, S: Simulation.

| Pig | CV (cm/s) | | LDC | APD$_{90}$ (ms) | | $g_{K1}$ |
|---|---|---|---|---|---|---|
| | E | S | (cm²/ms) | E | S | (%) |
| 8 | 32.8 | 31.7 | 0.001 | 346 | 345.3 | 58 |
| 9 | 27.9 | 26.9 | 0.00046 | 482 | 476 | 19 |
| 10 | 38.6 | 39.0 | 0.00095 | 409 | 406 | 31 |
| 11 | 31.1 | 31.8 | 0.0009 | 606 | 557 | 1 |
| 12 | 43.6 | 43.7 | 0.0011 | 361 | 354.5 | 36 |

for LDC values between 0.00046 and 0.0011 cm²/ms. For APD$_{90}$, differences for pigs 8–12 were 0.2%, 1.3%, 0.7%, 8%, and 1.8% when $g_{K1}$ was reduced from 1% to 58% of its default value. Based on these calibrations, BZ tissue was modeled with LDC = 0.000882 cm²/ms and $g_{K1}$ = 29% of baseline.

Simulated AT maps closely matched OM maps in the anterior BiV view (Fig 9), with activation initiating in the anterior RV, propagating to the LV, and terminating near the MI-affected LV apex. Maximum AT differences were 10 ms (pig 9) and 6 ms (pig 10). Experimentally, average CV decreased from 72.4 cm/s in HZ to 34.8 cm/s in BZ, while average $APD_{90}$ increased from 250 ms to 441 ms. Simulations yielded comparable values: CV = 87.3 / 34.6 cm/s in HZ / BZ and $APD_{90}$ = 247 / 428 ms in HZ / BZ.

Across pigs 8–12, simulated CVs were 20% lower (HZ) and 0.5% lower (BZ) than experimental values, with $APD_{90}$ differing by only 1% (HZ) and 3% (BZ).

### 3.6 Post-MI ventricular activation effect on the ECG

In G6 simulations, replacing ap-CS with mi-CS produced minimal changes in ventricular depolarization-repolarization dynamics. Fig 10a shows identical QRS complexes for both CS distributions across leads. Minor ventricular AT differences were restricted to the MI-affected LV free wall. The T wave remained unchanged, with only localized $APD_{90}$ alterations (see S3 Fig). Mean similarity to experimental ECG for pig 6 was low in both cases (QRS complex / T wave: 0.34 / 0.37 for ap-CS and 0.34 / 0.36 for mi-CS).

Comparison with EAM-based activation (Fig 10c) showed that e-EAM and i-EAM substantially improved T-wave similarity, particularly in leads aVF and V4 (Fig 10b and Fig 10c, right), with mean values increasing from 0.38 (ap-CS) to 0.52 (e-EAM) and 0.63 (i-EAM). However, QRS similarity decreased markedly (Fig 10c, left), with mean values of 0.34 (ap-CS), -0.11 (e-EAM), and 0 (i-EAM). Overall, EAM-based activation improved T-wave agreement but worsened QRS matching relative to experimental data.

### 3.7 *In silico* effect of MI morphology and location on the ECG

In G7 simulations, qualitative evaluation of ECGs under healthy and MI conditions was performed for pigs 4–12 across seven leads (Fig 11a). MI had minimal impact on the QRS complex but substantially altered the T wave.

For LCx MI, T-wave amplitude and duration increased in leads I, II, III, V2, and V5, with the largest changes in pigs 6–7, which had greater proportions of BZ within AZ than pigs 4–5: 20%, 21%, 32%, and 45.5% for pigs 4–7,

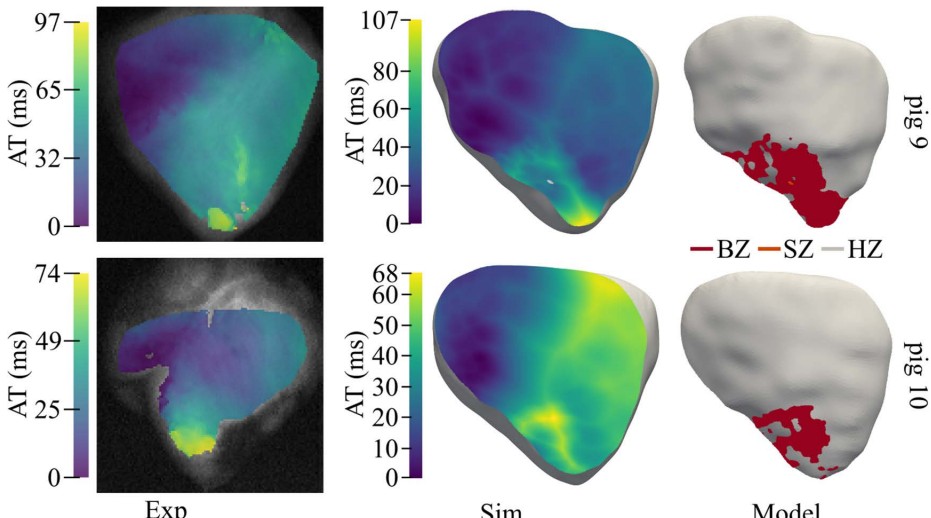

**Fig 9. Experimental and simulated activation after MI.** Experimental (left) and simulated (middle) AT maps of the anterior view for two MI pigs with LAD occlusion. The anterior view of the model for the two pigs is shown at the right side. AT: Activation time, MI: Myocardial Infarction, LAD: Left anterior descending, BZ: Border zone, SZ: Scar zone, HZ: Healthy zone.

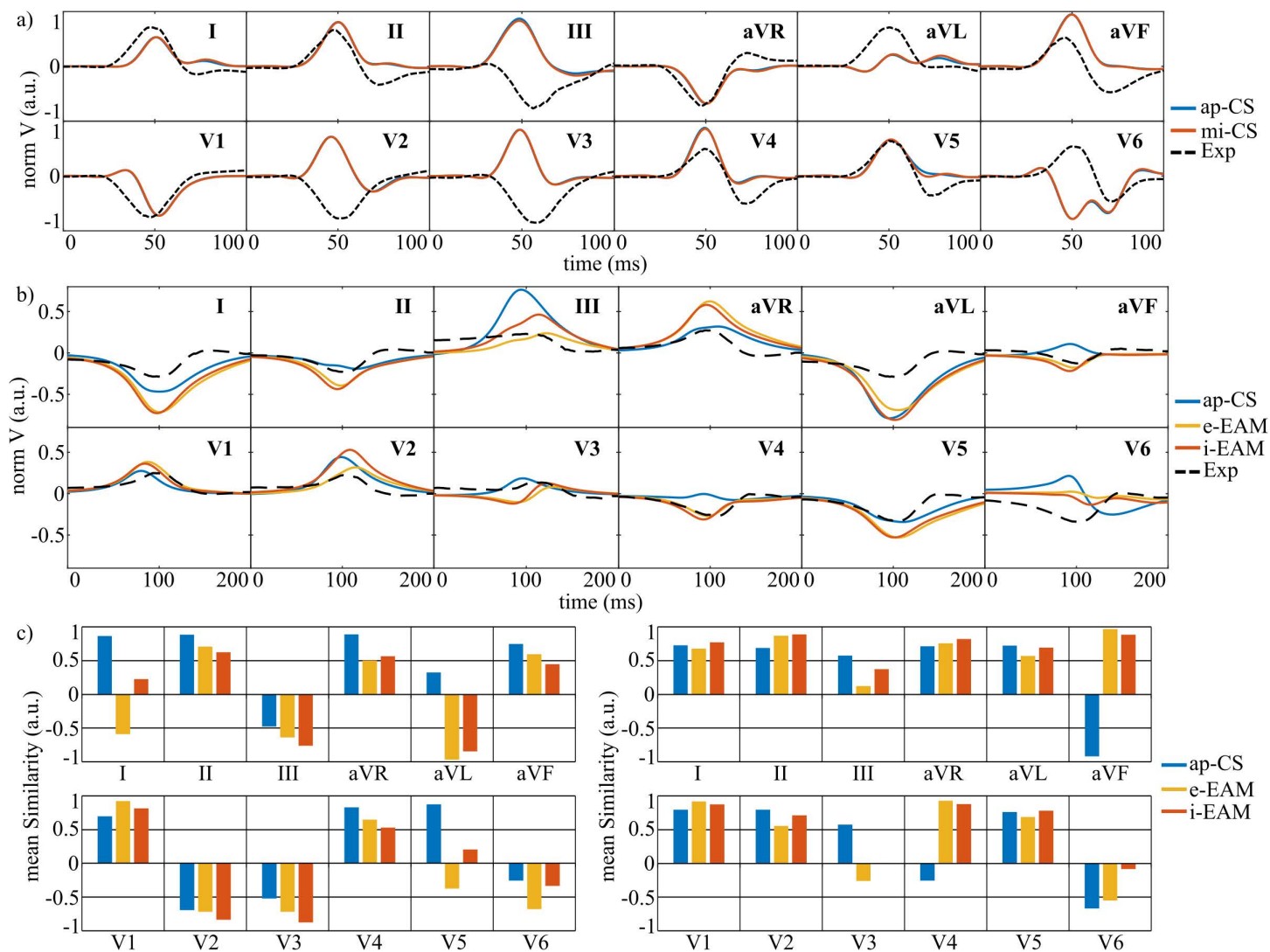

**Fig 10. Effect of post-MI altered activation on the ECG.** a- Experimental and simulated QRS complexes for pig 6 when ap-CS and mi-CS were used. b- Experimental and simulated T waves for pig 6 when the activation was generated by ap-CS and e-EAM and i-EAM stimulations. c- QRS (left) and T-wave (right) mean similarity for different activation approaches. CS: Conduction system, ap-CS: intramyocardial CS with Purkinje in the septal wall of the right ventricle and with the end of the left bundle branches closer to the apex of the left ventricle, mi-CS: ap-CS with altered spatial distribution of Purkinje fibers following infarction, e/i-EAM: Endocardial/intramyocardial activation based on the electroanatomical mapping.

respectively (Fig 11b). In these pigs, the QRS amplitude remained stable across all leads, with slight decreases in leads I and V5.

For LAD MI, the greatest T-wave amplitude increases occurred in leads III, V2, and V3, with T-wave inversion in leads III and V2 for pigs 8–10. The QRS complexes were more disrupted for LAD pigs than for LCx pigs. Leads presenting post-MI T-wave inversion showed a marked QRS distortion compared to the healthy case. Interestingly, a Q wave appeared in lead I for most LAD pigs.

Taken together, enlarged T waves in leads I, II, and V5 may indicate LCx MI, with greater enlargement suggesting a larger MI size. Enlarged T waves in lead V3 or large and inverted T waves in leads III and V2 may indicate LAD MI.

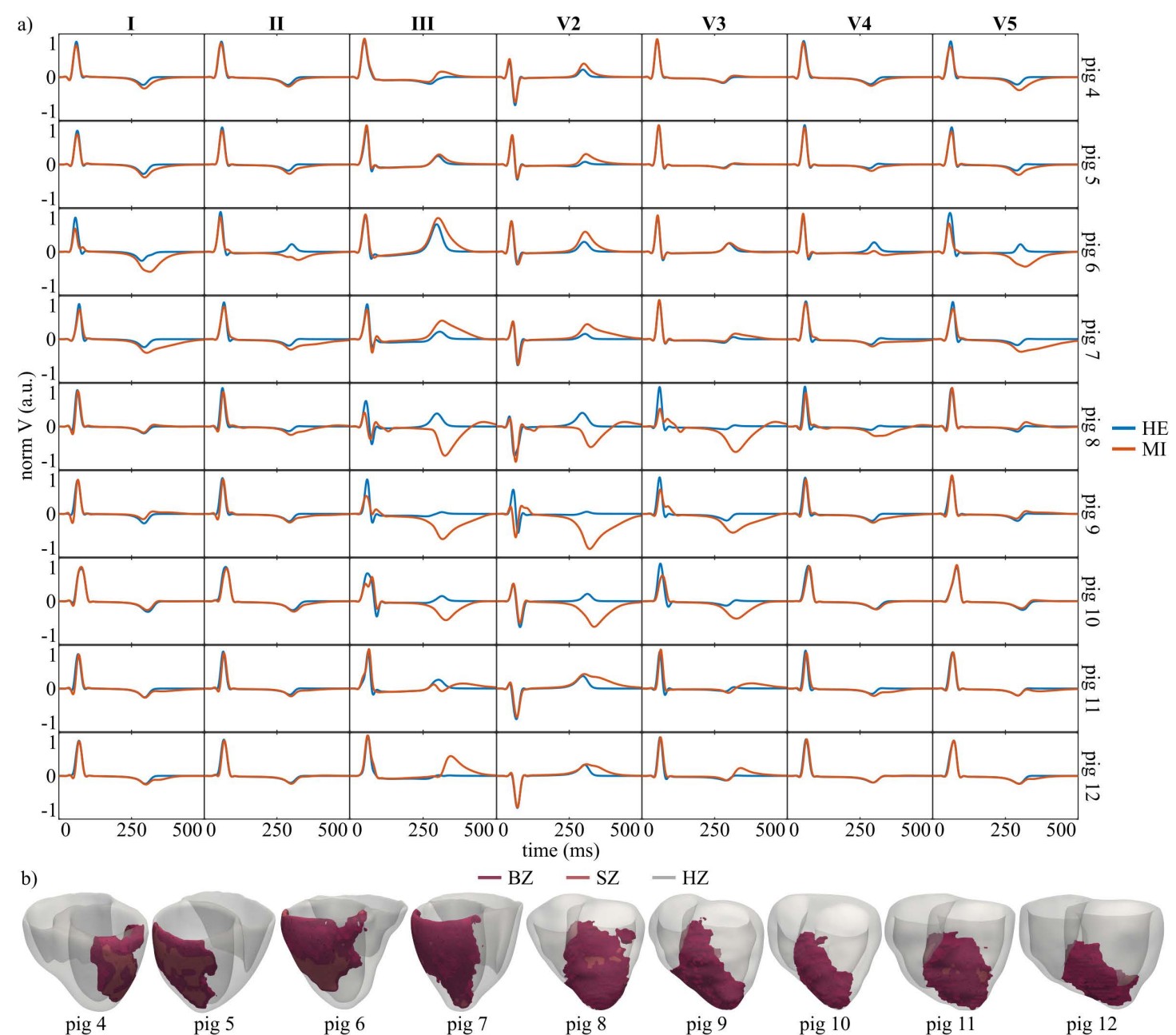

**Fig 11. ECG patterns and MI substrate specific to vessel-dependent occlusions.** a- Healthy and MI pECGs for pigs 4-12. b- MI BiV models for pigs 4-12 depicting the BZ and SZ. pECG: Simulated pseudo-electrocardiogram, MI: Myocardial infarction, BiV: Biventricular, HE: Healthy, HZ: Healthy zone, BZ: Border zone, SZ: Scar zone.

## 4 Discussion

We built swine-specific electrophysiological models that integrate extensive experimental data under both healthy and MI conditions. The main findings indicate that cardiomyocyte orientation exerted only a minor influence on cardiac electro-physiology, whereas CS architecture and APDSH were critical for reproducing normal depolarization and repolarization

patterns, respectively. In chronic MI, incorporating experimentally derived alterations in cell-to-cell coupling and APD into a swine cellular model enabled an accurate reproduction of the MI electrical phenotype. This experimentally guided approach also allowed us to replicate experimental T waves and characterize QRS and T-wave patterns as a function of MI location.

## 4.1 CS-myocardial coupling

A key component of our *in silico* models is the coupling between CS and the ventricular myocardium, which is fundamental to coordinated ventricular activation. Electrical depolarization is governed by the source-sink ratio, defined by the relationship between charge supplied by activated tissue (the source) and the charge required to activate the adjacent tissue (the sink) [63]. When this ratio falls below 1, conduction block occurs because the sink demands more current than the source can deliver. This mismatch is influenced by both structural and electrophysiological properties, including tissue bulk, cell count, intercellular coupling, and excitability [63]. In computational models, such source-sink mismatches have been shown to induce conduction block in diverse settings, from transmurally infarcted tissues remuscularized with engineered minitissue [34] to 3D models of sinoatrial node-atrial coupling [64].

A relatively unexplored source-sink mismatch arises in PMJ modeling, where thin Purkinje bundles (modeled as line elements) connect to bulkier myocardium (represented with tetrahedral meshes), as in the present study. Conduction block can occur at PMJs despite the higher excitability and stronger intercellular coupling of the CS compared to the BiV myocardium, both defined according to experimental observations. In addition to the larger BiV tissue volume, the high spatial resolution of many simulation meshes can exacerbate this block [11,20].

Our results revealed an optimal PMJ radius that minimizes activation block when all variables contributing to the source-sink mismatch were fixed. Increasing the PMJ radius generally increased the number of CS-myocardial connections, which initially boosted the number of activated PMJs. However, not all connected PMJs contributed to forward activation, as some were retrogradely activated from the myocardium (Table 5). The maximum number of myocardial activations from CS occurred once the PMJ radius allowed connection to a few dozen myocardial nodes, which is in line with a previous computational work where 50 connections per PMJ were employed for CS-myocardium coupling when a hexahedral BiV mesh with 0.5 mm edge length was used to represent the ventricular myocardium [52]. Beyond this, excessive PMJ-myocardium connections worsened the source-sink mismatch, reducing the number of PMJs with anterograde activation. These findings held for both endocardial and intramyocardial CS distributions, with intramyocardial CS showing the expected drop in the number of PMJs with anterograde activation at slightly smaller PMJ radii due to the greater number of connected myocardial nodes.

## 4.2 Porcine CS architecture and electrode positioning as the determinants of the QRS complex

The ventricular depolarization pattern initiated by CS is reflected in the QRS complex of the ECG. In this work, we simulated pECG signals to evaluate how QRS morphology is influenced by CS architecture in combination with HERP.

Three modifications to a baseline CS configuration were found to improve agreement between simulated (pigs 1–3) and experimental (pig 15) QRS complexes:

- *Addition of Purkinje fibers to the RV septum:* The baseline bi-CS configuration failed to reproduce the predominantly positive QRS complexes in leads I and II. Adding Purkinje fibers to the RV septum promoted right-to-left septal depolarization, producing more positive QRS morphologies in these leads and their augmented counterparts (aVR and aVL). This suggests that the RV septum is directly activated by the RV Purkinje branch rather than indirectly from LV activation propagating across the septum, consistent with some experimental reports of RV septal activation in human hearts [49].

- *Adjustment of HERP:* Rotation of the heart, combined with more central positioning of the precordial electrodes, improved the match between simulated and experimental QRS morphologies in precordial leads. In pig 15, leads V1-V5

showed similar morphology and predominantly negative R waves despite being positioned on opposite sides of the anterior thorax. The adjusted a-HERP increased RV exposure to precordial leads, enabling right-to-left depolarization to appear consistently in leads V1-V5 as a negative QRS deflection, matching experimental observations.

- *Apical repositioning of LV bundle branch endpoints:* Shifting the anterior and posterior LV bundle branch endpoints more apically enhanced apex-to-base depolarization, improving agreement in leads I and II and in augmented leads without disturbing precordial lead similarity. Additionally, this modification in the fast CS resulted in a more negative QRS complex in lead III, bringing it closer to the experimental morphology. Nonetheless, residual mismatches remained in leads II, III, and aVF, while lead V6 remained challenging to match as its position corresponds to the septal base, a region lacking in our model. This anatomical discrepancy makes V6 highly sensitive to even minor shifts in electrode or heart positioning.

Beyond these refinements, the depth of CS penetration emerged as a critical factor. Simulations with intramyocardial ap-CS distributions reproduced experimental QRS complexes better than their purely endocardial counterparts. This is consistent with anatomical studies reporting a deeper CS location in pigs than in humans [8]. While human models of QRS generation are numerous [48,52,56], few have validated simulated QRS morphology against experimental data across all leads for a full BiV model [21] or have used a topologically detailed CS [65,66]. Here, we provide CS and HERP definitions validated against experimental ECGs for three porcine-specific BiV models.

OM-based epicardial activation measurements also matched simulations. In sinus rhythm, activation started in the RV and ended in the basal LV for anterior and posterior OM views. The first epicardial breakthrough consistently occurred in the RV area pretrabecularis, matching human experimental observations [49,67]. This pattern likely arises from the thin RV wall and RV bundle branch geometry, which travels from the septal base to the anterior wall via the septomarginal trabecula, a structure well documented in mammals [68] and located higher, more basally, in pigs than in humans [8]. Simulated maximum AT and CV closely matched experimental values, though simulations yielded slightly higher median CVs (see Section 4.4).

Despite stochastic CS variations, the preservation of QRS morphology across pigs 1–3 (same breed, weight, and age) highlights the robustness and reliability of our modeling approach. In this context, a sensitivity analysis of non-species-specific parameters demonstrated that a ±20% variation in myocardial (S5 Fig) and CS (S6 Fig) conductivity resulted in, at most, a 10% change in maximum AT and CV, respectively. Notably, QRS complex morphology remained largely unaffected.

Overall, with appropriately defined CS architecture, HERP, and electrophysiological parameters, our healthy porcine models accurately replicate experimental QRS complexes and epicardial activation patterns.

## 4.3 Small APD heterogeneity defines porcine ventricular repolarization

The porcine ventricular cell model used here [10] captures the overall swine ventricular phenotype but does not include location-specific definitions of repolarization. As a result, initial simulations were performed without regional APD variation, consistent with our OM experiments and with findings from Kong et al. [69], who reported no significant $APD_{90}$ differences in swine ventricles. In contrast, Meijborg et al. [59] have found statistically significant spatial repolarization differences using regional repolarization (defined as the time when 50% of recording sites repolarize). Based on those findings, we implemented two APDSH configurations, t-APDSH and c-APDSH, for additional simulations.

The incorporation of APDSH improved the replication of the experimental T wave in leads II, III, and aVF for all three healthy cases, without altering the QRS complex. Lead V6 remained the most challenging for T-wave matching, likely due to the specific electrode position combined with the incomplete representation of the ventricular base and outflow tracts in the model. With c-APDSH, lower T-wave similarity was observed in leads I and aVR for pigs 2 and 3, but not for pig 1, suggesting inter-animal differences, which possibly make pig 1 a better surrogate for pig 15.

At the tissue level, simulated median $APD_{90}$ values on the epicardial surface matched OM-derived repolarization times. APDSH was not clearly detectable in OM data, likely due to the 2 ms acquisition resolution, which may be insufficient to capture subtle APD variations in the porcine epicardium. Using APDSH settings based on Meijborg et al. [59], simulations show a maximum epicardial $APD_{90}$ difference of 8 ms (at a 1000 ms cycle length) between the ventricular base and mid-ventricle.

The proportions of transmural and apicobasal regions were defined based on human-related reports [57,58] and empirical estimations, respectively. To assess how the limited availability of porcine-specific experimental data on the size of these regions might affect our results, we performed a sensitivity analysis of transmural and apicobasal APDSH distributions on *in silico* repolarization. As shown in S4 Fig, ±10% variations in layer sizes produced consistent T-wave morphologies and median $APD_{90}$ values, indicating that the model is robust to small deviations in these parameters.

Overall, our findings support the presence of ventricular APDSH in pigs and align with experimental evidence that porcine repolarization begins in the epicardium and ends in the endocardium [59,70]. The magnitude of transmural and apicobasal APD differences appears smaller in pigs than in human models [71,72], which are sometimes (inappropriately) used to represent porcine electrophysiology [9]. To our knowledge, this is the first computational study to reproduce experimentally observed porcine APDSH, achieved here by modifying $g_{K1}$ in the Gaur et al. [10] model. This *in silico* mechanistic insight suggests that $I_{K1}$ plays a more dominant role in modulating the porcine APD than other $K^+$-related currents typically emphasized in human models. This finding is consistent with experimental evidence showing that pigs exhibit high sensitivity to proarrhythmic effects induced by $I_{K1}$ blockade [73], while remaining largely insensitive to $I_{Kr}$ inhibition [74]. However, because quantitative porcine-specific data on regional ionic current densities remain limited, further experimental studies are warranted to determine whether $I_{K1}$ modulation more accurately reflects the underlying porcine repolarization gradients.

## 4.4 Minor role of subject-specific fiber orientation calibration in porcine ventricular electrophysiology

We evaluated the impact of using DW-based pig-specific fiber orientations versus RBM-based orientations in porcine BiV models. Since *ex vivo* DW-CMR geometries differed substantially from *in vivo* cine-based geometries, the RBM fibers were adjusted using DW-CMR data (s-RBM to dw-RBM, Section 2.6.4).

Adjusting RBM fibers to match DW data produced no significant changes in simulated pECGs or tissue-level electrophysiology. A small decrease in median CV was observed with dw-RBM, most notably for pig 1, where s-RBM and dw-RBM differed the most. Simulated CVs still slightly exceeded experimental CVs, consistent with earlier observations in both healthy and MI settings (Sections 4.2 and 3.5). In the anterior epicardial view, experimental CV in pig 13 closely matched simulated CV in pigs 1–3 using dw-RBM, suggesting that discrepancies arose mainly from lower posterior CVs in the experiment.

These findings align with prior studies showing minimal electrophysiological differences between DW- and RBM-derived fiber fields in dogs and rats [55,75]. While patient- or animal-specific fiber architectures may be relevant in complex scenarios, such as arrhythmia vulnerability, our results suggest that, in healthy porcine ventricles, fiber orientation exerts only a minimal effect on ECG morphology and on depolarization-repolarization-derived electrophysiological parameters.

## 4.5 MI-induced structural and electrophysiological remodeling in pigs

Experimental data revealed significant structural and electrophysiological remodeling in chronic MI, which informed our *in silico* models.

DW-CMR analysis showed reduced fractional anisotropy in AZ compared to HZ in pigs with LAD (pig 14) or LCx (pig 7) occlusions. This likely reflects MI-induced microstructural alterations, such as collagen deposition, fiber disarray, and myocyte loss, well documented in both acute and chronic MI [31].

OM experiments revealed a 50% reduction in CV and an almost 200 ms prolongation of $APD_{90}$ in AZ compared to HZ. CV slowing can result from reduced excitability and AP upstroke velocity, linked to decreased $I_{Na}$, as reported in canine BZ patch-clamp studies [54], as well as from structural remodeling such as fibrosis [76], myocyte disarray, and reduced inter-cellular coupling [77]. The observed $APD_{90}$ prolongation matches reports of reduced $K^+$ currents in canine BZ [78], which, in human ventricular models, extend APD [11,20].

Following methodologies established for human models [11,20], we reduced $I_{Kr}$ and $I_{Ks}$ conductances in the Gaur et al. [10] porcine model to 28.9% of their default values. This modification led to a relatively modest 21% increase in $APD_{90}$, whereas applying the same reduction to $I_{K1}$ resulted in a markedly larger 63% prolongation. Given the model's greater sensitivity to the inward rectifier current, APD prolongation in this study was therefore achieved by decreasing $g_{K1}$. Furthermore, the characterization of $I_{Na}$ blockade proved paramount (S7 Fig); 6% variations led to noticeable changes in S-wave morphology of simulated QRS complexes and a 10% variation in maximum BiV AT and median CV within the BZ. Due to the scarcity of porcine-specific data on post-MI ionic remodeling, excitability changes were modeled using canine patch-clamp observations of $I_{Na}$ [54]. In contrast, structural remodeling and conductivity changes were captured using experimental DW-CMR-based anisotropy from MI pigs 7 and 14. Finally, LDC was calibrated to achieve CV agreement with OM data within the BZ.

The resulting MI models reproduced experimental activation patterns: the earliest breakthrough in the RV area pre-trabecularis and activation ending in the LV anterior apex. Simulated values of AT, CV, and $APD_{90}$ at the healthy and MI regions closely matched experiments, with one exception: in pig 11's $APD_{90}$ (606 ms) within the BZ exceeds the model's maximum attainable value (557 ms, even with $g_{K1}$ reduced to 1%).

## 4.6 Replication of QRS and T-wave characteristics in infarcted porcine hearts

Comparing experimental and simulated pECGs from infarcted pigs highlighted key factors in MI modeling.

*BZ modeling*: Fibrosis was not incorporated into the BZ in this study. Prior *in silico* work in humans suggests that adding varying degrees of fibrosis produces minimal changes in QRS and T-wave morphology, causing only small ST-segment deviations [20].

*SZ modeling*: We modeled SZ as an insulator, consistent with previous computational studies [4,9,11,20]. Modeling SZ as passive tissue, as in Connolly et al. [3], can create unrealistically large potential gradients and ST deviations [20], requires the calibration of additional resistance and resting potential parameters, and forces the SZ to act as an electro-tonic sink. This sink effect shortens the APD in BZ, necessitating the definition of non-physiologically larger APD in the BZ to compensate.

*CS modeling in MI*: Although applying the ap-CS distribution from pig 15 to pig 6 did not replicate the QRS complex, highlighting the variability of CS geometry across animals, it emphasized the importance of individualized CS modeling. Implementing MI-specific modifications to CS (mi-CS) based on experimental data [5] altered AT and $APD_{90}$ locally. Although these modifications produced minimal visible changes in the QRS complexes, they are likely to have more dras-tic electrical effects, such as affecting unidirectional block formation and arrhythmia inducibility [20,79,80]. Therefore, a CS distribution that matches sinus rhythm ECG morphology may still be insufficient for *in silico* arrhythmic risk evaluation.

*EAM-based LV activation*: Replacing ap-CS-based LV activation with EAM-derived patterns surprisingly reduced QRS similarity in all leads. We attribute this to a mismatch: the RV was still activated by the ap-CS Purkinje network, while the LV followed EAM timings, as EAM data were only available for the LV. High-accuracy QRS reproduction in prior EAM-based MI simulations [20] required manual annotation refinement and full BiV endocardial and epicardial mapping. Despite QRS mismatches, EAM-based LV activation improved the replication of T-wave morphology in several leads (especially aVF and V4). Intramural EAM activation (i-EAM) slightly outperformed endocardial EAM (e-EAM), reinforcing the importance of representing the intramural CS position observed in pigs. Overall, these results show that RV activation has a strong influence on QRS morphology, while the modeling of MI-related T-wave changes may benefit from invasive mapping integration.

## 4.7 Mechanisms underlying MI-induced ECG changes

The MI BiV models developed here provided insight into the mechanisms driving ECG changes in infarcted hearts. The most prominent effects appeared in the T wave, likely reflecting increased repolarization dispersion [56,59,70], while QRS morphology was only minimally altered.

Post-MI ventricular activation remained broadly similar to the healthy state. Although BZ excitability was reduced, the high PMJ density in our CS distributions likely limited the impact on global activation. This held true whether the CS distribution was unchanged or modified for MI (Section 3.6).

Concerning repolarization, LCx MI produced T-wave enlargement in leads V2, V5, and bipolar limb leads, which are more sensitive to LV-RV voltage gradients than to apicobasal gradients. Pigs 6 and 7, with larger BZ volumes, showed the greatest T-wave changes, consistent with stronger $V_m$ gradients during repolarization.

In LAD MI, T-wave magnification appeared in leads III, V2, and V3, with clear inversion in pigs 8–10, but not in pigs 11 and 12. This variability may be driven by differences in BiV geometry, MI substrate, or CS configuration. Notably, pigs 8–10 showed larger QRS changes in the same leads when T-wave inversion occurred, suggesting that altered activation patterns could modulate repolarization dispersion, which aligned with the improved *in silico* post-MI T-wave replication when experimentally measured LV activation was used (Section 4.6).

These results demonstrate the value of *in silico* modeling for dissecting the mechanisms by which MI alters the ECG. In our models, QRS stability in the face of MI is attributable to robust CS-mediated activation, while T-wave changes reflect MI-driven alterations in repolarization gradients and their interaction with lead orientation.

## 4.8 Limitations, uncertainties, and future work

OM is a powerful technique to characterize depolarization-repolarization cardiac cycles, but it has inherent limitations. The 2D fluorescence image obtained at each time point represents photons originating not only from the epicardial surface but also from deeper myocardial layers. Due to photon scattering and the 2D nature of OM, multi-view imaging and advanced reconstruction algorithms are required for full 3D recovery of cardiac electrical activity [81]. In this work, OM videos from anterior and posterior heart views were used, restricting comparisons between simulated and experimental data to these surfaces. The temporal resolution was 2 ms, sufficient to capture most ventricular electrical dynamics. OM datasets were available for sinus rhythm and pacing from multiple sites, with pacing data informing steady-state CV and $APD_{90}$ estimates and sinus rhythm data characterizing AT and depolarization patterns for simulation comparison.

*Ex vivo* and *in vivo* CMR data were analyzed. Even with care to minimize deformation, *ex vivo* CMR showed a smaller LV lumen and thicker myocardium compared to *in vivo* cine-CMR at end-diastole. Therefore, BiV simulation geometries were derived from *in vivo* CMR, with fiber fields mapped from *ex vivo* to *in vivo* geometries. Future work could employ improved protocols to further reduce *ex vivo* deformation [82]. The electrophysiological impact of these geometrical shape changes remains unclear. Durrer et al. [49] reported preserved activation patterns but a higher CV when canine hearts were mapped *ex vivo* (Langendorff) versus *in situ*. In our study, mean *ex vivo* CV values from OM are lower than those reported for *in situ* porcine OM [83].

We compared a standard RBM (s-RBM) with an RBM informed by DW data (dw-RBM). Direct use of DW-CMR fiber fields was not possible, as anatomical models were based on *in vivo* cine-CMR. The RBM has its own limitations. RBM imposes a linear transmural helix angle ($\alpha$) variation and constrains fibers to lie parallel to the apicobasal-circumferential plane [55]. While these assumptions have minimal effects on electrophysiological simulations, they may limit the *in silico* reproduction of experimentally observed shear strain in LV mechanics [84].

Fibrosis was not explicitly modeled, although BZ conductivity was reduced. Previous works showed mixed effects. Campos et al. [80] found only non-sustained arrhythmias when fibrosis was combined with reduced $I_{Na}$ and BZ conductivity, suggesting $I_{Na}$ reduction, not fibrosis, as the key proarrhythmic factor. Lopez-Perez et al. [20] reported that fibrosis plus electrophysiological remodeling increased arrhythmic risk, whereas fibrosis alone had little effect on ECG or activation.

Individualizing fibrosis distribution requires invasive mapping, and myofibroblast-cardiomyocyte coupling mechanisms remain incompletely understood.

Uncertainties in model definition and the use of overly simplistic or inappropriate numerical settings can distort the results of *in silico* studies [9,11,85]. Sensitivity analyses of key parameters, provide a rigorous means of assessing these uncertainties but are computationally demanding and time-consuming. Although a detailed sensitivity analysis was beyond the scope of this study, we adhered to established best practices to maximize confidence in our results:

- *Segmentation delineation:* Mesh generation relied on state-of-the-art algorithms trained on large real-world datasets, with only minor adjustments applied for BiV modeling [25].

- *Mesh discretization:* Tetrahedralizations followed edge lengths consistent with previous studies [9,11,20,44], which have reliably captured both physiological and arrhythmic dynamics. In particular, Boyle et al. [86] demonstrated that edge lengths below 400 µm can resolve reentrant circuits consistent with experimental observations while avoiding numerical conduction block artifacts. More recently, Bishop et al. [85] highlighted the need for finer discretizations under low-CV conditions to prevent spurious conduction block and variability in arrhythmia outcomes. Although arrhythmicity was not directly assessed in this work, we argue that the discretization adopted here remains suitable for such contexts, consistent with Sung et al. [87]. Furthermore, we implemented Bishop et al.'s recommendations for CV tuning in coarser meshes [85]: CVs in infarcted regions were case-specifically adjusted using OM data when available, while conductivity settings in healthy zones produced CVs consistent with experimental values, particularly when comparing anterior-view measurements in pigs 1–3 to pig 13.

- *Cross-species data:* Although pigs are widely regarded as a robust translational model for human cardiac electrophysiology, relevant inter-species differences remain. In particular, quantitative porcine data of APDSH ratios, CS electrophysiology, and BZ ionic remodeling remain scarce. Consequently, several parameters were adapted from human or canine studies that reproduce qualitative electrophysiological behaviors characteristic of large mammalian hearts [11,20,49,50,51,53,54,57,58]. To assess the impact of this limitation, a sensitivity analysis was performed on APDSH ratios (S4 Fig), myocardial (S5 Fig) and CS (S6 Fig) conductivities, as well as post-MI $I_{Na}$ remodeling (S7 Fig). At the ECG level, only minor variations were observed, including slightly shorter or longer QRS complex durations for LDC variations in the CS and myocardium, and small S-wave deflections associated with increased BZ excitability remodeling. At the tissue level, electrophysiological markers were insensitive to variations in APDSH ratios. Modifying CS and myocardial LDC by ±20% changed maximum AT and median CV by less than 10%. In contrast, ±6% variations in BZ $g_{Na}$ resulted in changes of approximately 10% in activation-related metrics, underscoring the need for improved calibration of this parameter using porcine-specific post-MI data to further reduce output uncertainty. Overall, this sensitivity analysis suggests that most cross-species parameter choices do not compromise model robustness; however, certain parameters may introduce bias. To mitigate this effect, key outputs (AT, CV, $APD_{90}$, depolarization patterns, and ECG morphology) were calibrated against porcine experimental OM and ECG data, ensuring physiological consistency with the porcine phenotype.

- *Cross-subject data:* Not all experimental modalities were available here for each animal under both healthy and MI conditions, partly because some methods are incompatible (e.g., *ex vivo* DW-CMR versus OM) or center-specific (LAD-MI and OM were performed at SERMAS, and LCx-MI and EAM were conducted at KUL). Consequently, the extent of model individualization differed among subjects. When subject-specific data were unavailable, population-average values were used. This strategy is common in high-fidelity *in silico* studies, given the complexity and cost of full multi-modal datasets [16–18].

- *Beat-to-beat variability:* Our simulations demonstrated strong agreement with the median OM findings rather than on a per-beat basis, as the experimental results exhibited inherent beat-to-beat variability that was not captured by our

deterministic modeling framework. Future studies could incorporate stochastic modeling to better reproduce the variability observed in the experimental data, which was particularly prominent in specific regions and samples (see Section 2.3).

- *Modeling complexity:* While fine-grained, local variations in APD were used to reproduce experimental ECG and OM data, global conductivities were applied to replicate experimental activation features in both healthy and MI conditions. Previous research stemming from the CESC10 [16] and CRT-EPiggy19 [17] challenges has underscored the necessity of regional CV modeling to improve *in silico* agreement with experimental results. Specifically, Camara et al. [19] demonstrated that phenomenological-Eikonal modeling, when combined with detailed biophysical models, enhances the reproduction of OM-measured paced activation in porcine models using DW-CMR-defined geometries and fiber fields. Furthermore, Albors et al. [22] improved the prediction of EAM activation in left-bundle branch block conditions by partitioning ventricular tissue into regions and optimizing conductivity for each of these regions. Despite our use of global conductivities, our approach successfully replicated depolarization characteristics from both tissue-level OM data and organ-level relevant ECG measurements.

Future work should leverage more granular modeling pipelines that integrate findings from previous computational challenges with novel, species-specific experiments, which are currently lacking in the literature, to further validate and extend our observations while reducing reliance on cross-species and / or cross-individuals parameterization. Despite the mentioned constraints, this study integrates an exceptionally broad range of experimental data. It extends the efforts of previous benchmarks, such as CESC10 [16] and CRT-EPiggy19 [17], by incorporating multi-sequence functional and structural CMR, OM and EAM measurements in sinus rhythm, and clinically relevant ECG data. Ultimately, this integrated pipeline identifies the modeling features critical for replicating ventricular electrophysiology across multiple scales in both healthy and infarcted porcine hearts.

## 4.9 Model usability and preclinical implications

The developed models provide a high-fidelity *in silico* platform to investigate how healthy and MI conditions affect electrophysiological outcomes (ECG and OM) in a translational porcine model. Beyond mechanistic insights, these models may support preclinical cardiovascular applications, including the assessment of drug proarrhythmicity [10], cardiac resynchronization therapy [17], and the optimization of cardiac tissue engineering strategies. Specifically, our framework can inform the design and implantation of engineered heart tissues for remuscularization [34,88].

Given that post-MI remuscularization is currently in the preclinical clearance phase [89–91], these models constitute valuable tools for assessing arrhythmic risk [13,88], thereby reducing reliance on animal experimentation. To ensure suitability for this purpose, we employed a multi-modal validation strategy with predefined accuracy thresholds:

- *Depolarization and repolarization:* Accuracy was evaluated at the tissue level by comparing simulated maximum ATs, mean CVs, and mean $APD_{90}$ values with OM data. Acceptable tolerances were defined based on experimental uncertainty and established benchmarks. For AT, we targeted a difference of ≤ 10 ms [19,49,67], while simulated CV and $APD_{90}$ values were required to lie within the experimental mean ± standard deviation.

- *Activation patterns:* A qualitative agreement with experimental depolarization patterns was required, particularly during sinus rhythm where activation is not externally paced [9,16,19]. Benchmarks included the first epicardial breakthrough occurring in the RV area pretrabecularis and the latest activation in the posterobasal LV/RV regions [49,67].

- *ECG morphology:* At the organ level, consistency in wave polarity, morphology, and duration was prioritized. We targeted a Pearson correlation coefficient ≥ 0.7 between experimental and simulated QRS complexes and T waves [66].

Our results show that the models successfully met the predefined thresholds whenever experimental data were available for comparison. Two specific exceptions were identified: simulated CV was higher only in the posterior view of the

heart (G4 simulations), and QRS similarity was lower for pig 6 (G6 simulations), as the CS was defined using an ECG from a different subject. These discrepancies may affect the accuracy of arrhythmogenesis predictions, underscoring the need for regional CV calibration [22] and subject-specific experimental data to enable more precise digital twinning. The latter is supported by the observation that T-wave similarity nearly reached the target threshold once post-MI LV activation alterations were implemented. Overall, our cohort provides a physiologically consistent representation of the porcine phenotype under both healthy and MI conditions, offering a robust foundation for evaluating arrhythmicity through *in silico* inducibility protocols and other translational applications.

## 5 Conclusion

We develop and validate *in silico* porcine ventricular models for both healthy and MI conditions, using extensive multi-modal experimental data. In healthy ventricular models, precise CS architecture and realistic APD spatial heterogeneity are essential to match simulated and measured ventricular activation (OM) and ECG morphology. In MI models, incorporating altered tissue properties and $K^+$ conductances reproduce key MI-induced electrophysiological changes, notably prominent T-wave alterations following LAD and LCx occlusions. Simulation analysis reveal that these T-wave changes arise from MI-induced repolarization gradients, while QRS stability reflects robust CS-mediated activation. This framework provides a reproducible data-driven pathway for building and validating individualized porcine electrophysiological models, enabling mechanistic dissection of healthy and pathological ventricular ECG features.

## Supporting information

**S1 Fig. DW-CMR acquisition protocols.** Representative transverse view of the healthy porcine heart (pig 3) acquired at KUL (left) and coronal view of the LAD-occlusion MI model (pig 14) acquired at SERMAS (right). DW-CMR: Diffusion-weighted cardiac magnetic resonance, KUL: Katholieke Universiteit Leuven, SERMAS: Servicio Madrileño de Salud, LAD: Left anterior descending, MI: Myocardial infarction.
(TIFF)

**S2 Fig. Simulated and experimental ECGs under healthy conditions.** The complete simulated ECGs obtained for the three healthy pigs 1–3 when using the adjusted CS, APDSH, and s-RBM are shown together with the experimental ECG of pig 15. ECG: Electrocardiogram, CS: Conduction system, APDSH: Action potential duration spatial heterogeneities, s-RBM: Standard rule-based model.
(TIFF)

**S3 Fig. Effect of post-MI altered repolarization.** a- Experimental and simulated T waves for pig 6 when ap-CS and mi-CS were used. b- Simulated $APD_{90}$ maps for pig 6 when the activation was generated by ap-CS and mi-CS. MI: Myocardial infarction, ap-CS: Intramyocardial, porcine conduction system with Purkinje in the septal wall of the right ventricle and the end of the left bundle branches closer to the apex of the left ventricle, mi-CS: ap-CS with altered spatial distribution of Purkinje fibers following infarction, $APD_{90}$: Action potential duration at 90% repolarization.
(TIFF)

**S4 Fig. Sensitivity analysis of APDSH distribution on the simulated ECG.** a- Twelve APDSH configurations were evaluated against the baseline APDSH, which was defined using transmural and longitudinal ratios adapted from non-porcine sources. Specifically, APDSH variations were generated by decreasing the size of one layer by 10% while simultaneously increasing the remaining two layers by 5% each; this was performed independently for both transmural and longitudinal directions. b- Maximum AT and median $APD_{90}$ and CV values obtained at the anterior view of models with different APDSH configurations when simulated with a 1000 ms cycle length. c- Simulated ECGs demonstrating no variation in QRS complexes and negligible alterations in T waves across all tested APDSH ratios. d- Magnified simulated T-waves

highlighting the preservation of T-wave morphology, with only minimal amplitude fluctuations primarily observed in leads II, III, aVF, and V4–V6. APDSH: Action potential duration spatial heterogeneities, ECG: Electrocardiogram, AT: Activation time, $APD_{90}$: Action potential duration at 90% repolarization, CV: Conduction velocity.
(TIFF)

**S5 Fig. Sensitivity analysis of ventricular conductivity.** The baseline LDC of the healthy myocardium was varied by ±20% in pig 1. a- Complete simulated ECG signals. b- Magnified simulated QRS complexes. c- Simulated maximum AT and median $APD_{90}$ and CV values obtained at the anterior face of the BiV model of pig 1. Simulated ECG and electrophysiological outputs were obtained under *in silico* pacing with 769 ms and 1000 ms cycle length, respectively. LDC: Longitudinal diffusion coefficient, ECG: Electrocardiogram, AT: Activation time, $APD_{90}$: Action potential duration at 90% repolarization, CV: Conduction velocity, BiV: Biventricular.
(TIFF)

**S6 Fig. Sensitivity analysis of CS conductivity.** The baseline LDC of the CS was varied by ±20% in pig 1. a- Complete simulated ECG signals. b- Magnified simulated QRS complexes. c- Simulated maximum AT and median $APD_{90}$ and CV values obtained at the anterior face of the BiV model of pig 1. Simulated ECG and electrophysiological outputs were obtained under *in silico* pacing with 769 ms and 1000 ms cycle length, respectively. CS: Conduction system, LDC: Longitudinal diffusion coefficient, ECG: Electrocardiogram, AT: Activation time, $APD_{90}$: Action potential duration at 90% repolarization, CV: Conduction velocity, BiV: Biventricular.
(TIFF)

**S7 Fig. Sensitivity analysis of BZ $I_{Na}$ remodeling.** The $g_{Na}$ value applied to the BZ (38% of the healthy, default value) was varied by ±6% (32% and 44% of the healthy, default value) in pig 6. a- Complete simulated ECG signals. b- Magnified simulated QRS complexes. c- Simulated maximum ventricular AT and median $APD_{90}$ and CV values within the BZ. Simulated ECG and electrophysiological outputs were both obtained under *in silico* pacing with 1000 ms cycle length. BZ: Border zone, $I_{Na}$: Sodium current, $g_{Na}$: Maximum conductance of the $I_{Na}$, ECG: Electrocardiogram, AT: Activation time, $APD_{90}$: Action potential duration at 90% repolarization, CV: Conduction velocity.
(TIFF)

## Acknowledgments

Authors thank Ana María Sánchez de la Nava from SERMAS for acquiring the OM data and Dylan Vermoortele from KUL for providing the healthy porcine ECG. Computations were performed using ICTS NANBIOSIS (HPC Unit at University of Zaragoza).

## Author contributions

**Conceptualization:** Ricardo Maximiliano Rosales, Ana Mincholé, Esther Pueyo.

**Data curation:** Ricardo Maximiliano Rosales, Ming Wu, Piet Claus, Stefan Janssens, Gonzalo R. Ríos-Muñoz, María Eugenia Fernández-Santos, Pablo Martínez-Legazpi, Javier Bermejo, Aiden Flanagan.

**Formal analysis:** Ricardo Maximiliano Rosales.

**Funding acquisition:** Manuel Doblaré, Ana Mincholé, Esther Pueyo.

**Investigation:** Ricardo Maximiliano Rosales.

**Methodology:** Ricardo Maximiliano Rosales, Esther Pueyo.

**Project administration:** Esther Pueyo.

**Resources:** Manuel Doblaré, Ana Mincholé, Esther Pueyo.

**Software:** Ricardo Maximiliano Rosales, Ana Mincholé.

**Supervision:** Manuel Doblaré, Ana Mincholé, Esther Pueyo.

**Validation:** Ricardo Maximiliano Rosales.

**Visualization:** Ricardo Maximiliano Rosales.

**Writing – original draft:** Ricardo Maximiliano Rosales.

**Writing – review & editing:** Ricardo Maximiliano Rosales, Ming Wu, Piet Claus, Stefan Janssens, Gonzalo R. Ríos-Muñoz, María Eugenia Fernández-Santos, Pablo Martínez-Legazpi, Javier Bermejo, Aiden Flanagan, Manuel Doblaré, Ana Mincholé, Esther Pueyo.

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
