## [Decision Letter · Decision Letter 0]

3 Dec 2025

PCOMPBIOL-D-25-02241

Integrated multi-modal data analysis for computational modeling of healthy and location-dependent myocardial infarction conditions in porcine hearts

PLOS Computational Biology

Dear Dr. Rosales,

Thank you for submitting your manuscript to PLOS Computational Biology. After careful consideration, we feel that it has merit but does not fully meet PLOS Computational Biology's publication criteria as it currently stands. Therefore, we invite you to submit a revised version of the manuscript that addresses the points raised during the review process.

We look forward to receiving your revised manuscript.

Kind regards,

Andrew D. McCulloch, Ph.D.

Academic Editor

PLOS Computational Biology

Marc Birtwistle

Section Editor

PLOS Computational Biology

**Journal Requirements:**

2) We noticed that you used the phrase 'data not shown' in the manuscript. We do not allow these references, as the PLOS data access policy requires that all data be either published with the manuscript or made available in a publicly accessible database. Please amend the supplementary material to include the referenced data or remove the references.

Potential Copyright Issues:

- Please confirm (a) that you are the photographer of Figure 1, or (b) provide written permission from the photographer to publish the photo(s) under our CC BY 4.0 license.

- Figure 1. Please confirm whether you drew the images / clip-art within the figure panels by hand. If you did not draw the images, please provide (a) a link to the source of the images or icons and their license / terms of use; or (b) written permission from the copyright holder to publish the images or icons under our CC BY 4.0 license. Alternatively, you may replace the images with open source alternatives. See these open source resources you may use to replace images / clip-art:

5) Please ensure that the funders and grant numbers match between the Financial Disclosure field and the Funding Information tab in your submission form. Note that the funders must be provided in the same order in both places as well.

State the initials, alongside each funding source, of each author to receive each grant. For example: "This work was supported by the National Institutes of Health (####### to AM; ###### to CJ) and the National Science Foundation (###### to AM).".

**Reviewers' comments:**

Reviewer's Responses to Questions

**Comments to the Authors:**

Reviewer #1: Rosales et al. build sophisticated computer models of pig cardiac electrophysiology based on rich experimental data sets including myocardial infarction.

They then study the sensitivity of selected simulation results (activation times, conduction velocity, ECG) on specific model parameters. Some of them are tuned then and a few comparisons to measured data are shown for validation purposes. The manuscript is very extensive; in general well written and supported by helpful figures an tables.

My main comments are as follows:

The added value of this study could be carved out more clearly. The three specific goals mentioned are not fully adequate in my opinion

Goal 1: Why "to guide the development"? In my view you actually develop such models in the present study.

Goal 2: porcine-specific (in general) or specific to an individual pig?

Goal 3: "clinical application" is an overstatement in my opinion as most readers would likely assume application to humans

The added value of the developed models remains vaguely described. There is mention of supporting the 3R principles but no concrete ways in which this could be achived are mentioned.

I am surprised that there is little ST deviation (elevation or depression) visibile in the infarction ECGs (Fig. 11).

l. 190: How much beat-to-beat variability did you observe in the activation and APD90 values?

l. 322: "These porcine-specific pig-wise values of LDC and gK1 defined the mean values that were used in cases where personalization was not possible due to the lack of electrophysiological data."

Unclear to me

It is a pity that ECG is only available for two of the pigs. This would be a very valuable resource for model validation.

Does the need for separate alignment of QRS complexes and T-waves indicate thate AP duration / morphology is not well matched between simulations and experiments?

Fig. 7: What is the reason for V6 being negative in all simulations whereas it is positive in the experimental recording?

The data acquisition and processing for this study is a major effort. How were the cine CMR data described in section 2.2.1 used? I have the impression that only the end-diastolic state was used.

Code and model setups are available, which is comendable. What about the raw data? It seems that a huge part of the effort is related to data acquisition and not all of them were used, so it would be a pity if others couldn't build on it.

The manuscript makes use of so many abbreviations that it is hard to remember all of them, which makes following the results section tedious, e.g. "At small PMJ radii as 0.1 mm, ncPMJ was low (28.5% for be-CS, 10.4% for bi-CS), resulting in limited aPMJ (28% for be-CS, 7% for bi-CS)."

The same applies to many figure captions, which are full of abbreviations preventing the reader to interpret the figure without having found and memorized all abbreviations. Figure 10 for example is uninterpretable to me without a list of abbreviations (in mind).

Further comments:

I would advise to avoid the term "personalized" when referring to pigs rather than persons. "Individualized" might be an alternative.

The Cardiac Electrophysiological Simulation Challenge (CESC10); Camara et al., Prog Biophys Mol Bio 2011 and the CRT-EPIGGY19 https://crt-epiggy19.surge.sh seem worthwhile discussing in the context of this work.

l. 140: What are "custom-designed ad hoc containers"?

Why did you use the multi-diffusion approach to differentiate the different transmural layer and not also a Laplace solution as for the apico-basal gradient?

l. 380: Is there experimental evidence that spatial APD heterogeneity in pigs is due to differential expression of IK1?

Table 3 misses units

l. 392: "80-µA/cm2" -> "80 µA/cm2"?

l. 398: I thought optical mapping data were used to parametrize APD, how can it be used for validation then?

Section 3.5 also mentions that optical mapping data were used to calibrate border zone electrophysiology. Please clarify.

Fig. 6b: It is not obvious at first sight why there is only 1 "exp" trace if there are 3 pigs. From Table 1 it becomes clear that this is not a 1:1 comparison. This should be highlighted in the figure caption.

Fig. 9 should clearly indicate the scar region (which should not be activated at all as I understand).

"Personalized fiber orientation" can be misleading as you did not personalize the entire fiber field but only one endocardial and one epicardial angle per ventricle as I understand.

Reviewer #2: Reviewer Comments

General Evaluation

This manuscript presents an ambitious whole-organ porcine digital-twin electrophysiology framework informed by high-quality imaging and electrophysiological measurements. The integration of fiber architecture, infarct structure, repolarization heterogeneity, and anatomical conduction system (CS) modeling is a strength, and the authors have assembled an exceptional experimental dataset from intact organs.

However, several methodological details require clarification, and some modelling assumptions rely on human or canine literature rather than porcine-specific measurements. The manuscript would benefit from clearer justification of those modelling choices, additional methodological transparency, and discussion of model validation requirements for future clinical translation.

Specific Comments

Abstract

• The similarity metric reported as:

“Accurate CS and repolarization heterogeneities reproduced depolarization (0.76 QRS similarity) and repolarization (0.74 T-wave similarity) patterns.”

should specify the calculation method (Pearson correlation). Without this information, the interpretation of the similarity values is unclear.

Scope and Framing of the Study

• The current framing presents the work largely as a modelling workflow rather than addressing either:

o a biological hypothesis, or

o an engineering performance target.

As written, the contribution sits between those categories. Clarifying the intended purpose of the model (mechanistic exploration, validation framework, clinical translation tool, or proof-of-feasibility) would help position the work appropriately.

Data

• Table 1 presents highly heterogeneous sample characteristics. The authors should comment on whether this variability was expected, and whether it impacts personalization steps or parameter calibration. It was not clear to me how data was combined (or not) to form a digital twin of a specific pig. Or is this more a model of a representative pig?

• Anatomical Model

• The mesh excludes the base of the heart. As the base is often the last region activated, it is unclear how this omission may affect the predicted QRS duration or conduction velocity calibration. Clarifying whether conduction was artificially slowed to compensate, or whether truncation explains QRS morphology mismatches, would strengthen the modelling justification.

Segmentation and Image Processing

• The statement:

“In MI cases, AZ and HZ were manually identified from the automatic deep learning–based segmentations…”

appears contradictory. It would help to clarify whether the method is:

o fully manual,

o fully automated, or

o automated with manual correction.

Electrophysiology Model Construction

A number of model components originate from non-porcine literature:

Component Source species

Sodium channel (gNa) remodeling Human [11,16] and canine [48]

Stewart Purkinje model Human [44]

Conduction system conductivity Human estimates [43]

Border zone remodeling Human and canine

Transmural repolarization gradients Human [51,52]

Because these choices may influence behaviour, the authors should comment on:

• expected inter-species electrophysiological differences,

• whether porcine-specific measurements are planned or needed,

• and whether substitution of human/canine parameters may bias conduction or repolarization dynamics.

APD Heterogeneity Strategy

• A sensitivity analysis on the imposed apex–base and transmural heterogeneity would be valuable, particularly because some transitions are implemented as discrete step changes rather than smooth gradients. It is unclear whether such abrupt transitions are physiologically plausible.

• Prior work (e.g., Camps J et al. Medical Image Analysis) implemented APD heterogeneity using IKs modulation in human models. Qian Nat Card Res used both Iks and Ikr in 3000+ cases. Gillete used Ikr.

• Discussion is warranted on whether:

o Kgr/gks modulation reflects species differences,

o a difference in underlying ion-channel experimental evidence,

o or a constraint of the specific porcine cellular model used.

Model Interpretation and Reporting

• Table 4 is difficult to interpret because several abbreviations and labels (e.g., s, b, a) are not defined. This prevents clear understanding of how spatial personalization was applied.

• The statement that “74 simulations were performed” requires clarification. Is this total across all pigs, or per animal?

Similarity Metric

• The Pearson correlation coefficient is a parametric measure. If used as a statistical inference tool, normality should be tested, or a non-parametric alternative considered.

• If it is intended purely as a signal similarity score, this should be stated explicitly and not framed as a hypothesis-testing metric.

Model Validation and Tolerances

• The manuscript would benefit from explicitly defining a use case for the model and the corresponding:

o how accuracy is assessed,

o what constitutes acceptable error for intended use, and

o whether the model meets those thresholds.

• The authors should include a sensitivity analysis, particularly in light of the non-porcine input parameters.

**Have the authors made all data and (if applicable) computational code underlying the findings in their manuscript fully available?**

Reviewer #1: **No:** No raw data available

Reviewer #2: **No:** There is no reason not to make the images, ECG and meshes available. I did not see a data link

PLOS authors have the option to publish the peer review history of their article (what does this mean? ). If published, this will include your full peer review and any attached files.

**Do you want your identity to be public for this peer review?** For information about this choice, including consent withdrawal, please see our Privacy Policy .

Reviewer #1: No

Reviewer #2: No

**Figure resubmission:**
---

## [Decision Letter · Decision Letter 1]

10 Feb 2026

PCOMPBIOL-D-25-02241R1

Integrated multi-modal data analysis for computational modeling of healthy and location-dependent myocardial infarction conditions in porcine hearts

PLOS Computational Biology

Dear Dr. Rosales,

Thank you for submitting your manuscript to PLOS Computational Biology. After careful consideration, we feel that it has merit but does not fully meet PLOS Computational Biology's publication criteria as it currently stands. Therefore, we invite you to submit a revised version of the manuscript that addresses the points raised during the review process.

We look forward to receiving your revised manuscript.

Kind regards,

Marc Birtwistle

Section Editor

PLOS Computational Biology

**Additional Editor Comments:**

Please confirm the materials are available in zenodo, reinforce the biological understanding that is gleaned and can be obtained using this computational advance (that was not before possible), and attend to the remaining concern of Reviewer 1.

**Reviewers' comments:**

Reviewer's Responses to Questions

**Comments to the Authors:**

Reviewer #1: Thank your for addressing my points and improving the manuscript.

I have one remaining point concerning beat-to-beat variability (#1 in my initial review):

I think it would be important to mention the experimental beat-to-beat variability that remains after quite strict beat selection (median standard deviation across pixels: 20.5ms for activation and 51.5ms for APD90 in pig 13) to i) put the remaining deviation during parameter tuning into perspective (doesn't make sense to aim for sub-milisecond precision during tuning when measurement variability/uncertainty is in the range of dozens of miliseconds); ii) highlight that while some of the observed variability is probably due to measurement uncertainty, there is very likely a contribution of intrinsic beat-to-beat variability as well, which is not considered in the model.

Reviewer #2: My concerns have largely been addressed. The manuscript would be further strengthened by more clearly articulating the specific biological question(s) that this modeling framework is intended to address.

**Have the authors made all data and (if applicable) computational code underlying the findings in their manuscript fully available?**

Reviewer #1: Yes

Reviewer #2: Yes

PLOS authors have the option to publish the peer review history of their article (what does this mean? ). If published, this will include your full peer review and any attached files.

**Do you want your identity to be public for this peer review?** For information about this choice, including consent withdrawal, please see our Privacy Policy .

Reviewer #1: No

Reviewer #2: No

**Figure resubmission:**
---

## [Decision Letter · Decision Letter 2]

16 Feb 2026

Dear Mr Rosales,

We are pleased to inform you that your manuscript 'Integrated multi-modal data analysis for computational modeling of healthy and location-dependent myocardial infarction conditions in porcine hearts' has been provisionally accepted for publication in PLOS Computational Biology.

Best regards,

Marc Birtwistle

Section Editor

PLOS Computational Biology

Reviewer's Responses to Questions

**Comments to the Authors:**

Reviewer #1: My remaining comments has been addressed. Congratulations on this study!

**Have the authors made all data and (if applicable) computational code underlying the findings in their manuscript fully available?**

Reviewer #1: Yes

PLOS authors have the option to publish the peer review history of their article (what does this mean? ). If published, this will include your full peer review and any attached files.

**Do you want your identity to be public for this peer review?** For information about this choice, including consent withdrawal, please see our Privacy Policy .

Reviewer #1: No

---

## [Editor Report · Acceptance letter]

PCOMPBIOL-D-25-02241R2

Integrated multi-modal data analysis for computational modeling of healthy and location-dependent myocardial infarction conditions in porcine hearts

Dear Dr Rosales,

I am pleased to inform you that your manuscript has been formally accepted for publication in PLOS Computational Biology. Your manuscript is now with our production department and you will be notified of the publication date in due course.

With kind regards,

Anita Estes
